# PROGRAM GUIDED AGENT

**Shao-Hua Sun, Te-Lin Wu, Joseph J. Lim**
University of Southern California
`{shaohuas,telinwu,limjj}@usc.edu`

## ABSTRACT

Developing agents that can learn to follow natural language instructions has been an emerging research direction. While being accessible and flexible, natural language instructions can sometimes be ambiguous even to humans. To address this, we propose to utilize programs, structured in a formal language, as a precise and expressive way to specify tasks. We then devise a modular framework that learns to perform a task specified by a program – as different circumstances give rise to diverse ways to accomplish the task, our framework can perceive which circumstance it is currently under, and instruct a multitask policy accordingly to fulfill each subtask of the overall task. Experimental results on a 2D Minecraft environment not only demonstrate that the proposed framework learns to reliably accomplish program instructions and achieves zero-shot generalization to more complex instructions but also verify the efficiency of the proposed modulation mechanism for learning the multitask policy. We also conduct an analysis comparing various models which learn from programs and natural language instructions in an end-to-end fashion.

## 1 INTRODUCTION

Humans are capable of leveraging instructions to accomplish complex tasks. A comprehensive instruction usually comprises a set of descriptions detailing a variety of situations and the corresponding subtasks that are required to be fulfilled. To accomplish a task, we can leverage instructions to estimate the progress, recognize the current state, and perform corresponding actions. For example, to make a gourmet dish, we can follow recipes and procedurally create the desired dish by recognizing what ingredients and tools are missing, what alternatives are available, and what corresponding preparations are required. With sufficient practice, we can improve our ability to perceive (*e.g.* knowing when food is well-cooked) as well as master cooking skills (*e.g.* cutting food into same-sized pieces), and eventually accomplish difficult recipes.

Can machines likewise learn to follow and exploit comprehensive instructions like humans? Utilizing expert demonstrations to instruct agents has been widely studied in (Finn et al., 2017; Yu et al., 2018b; Xu et al., 2018; Pathak et al., 2018; Stadie et al., 2017; Duan et al., 2017; Wang et al., 2017b). However, demonstrations could be expensive to obtain and are less flexible (*e.g.* altering subtask orders in demonstrations is nontrivial). On the other hand, natural language instructions are flexible and expressive (Malmaud et al., 2014; Jermsurawong & Habash, 2015; Kiddon et al., 2015; Misra et al., 2018; Fried et al., 2018; Kaplan et al., 2017; Bahdanau et al., 2019). Yet, language has the caveat of being ambiguous even to humans, due to its lacking of structure as well as unclear coreferences and entities. Andreas et al. (2017a); Oh et al. (2017) investigate a hierarchical approach, where the instructions consist of a set of symbolically represented subtasks. Nonetheless, those instructions are not a function of states (*i.e.* describe a variety of circumstances and the corresponding desired subtasks), which substantially limits their expressiveness.

We propose to utilize *programs*, written in a formal language, as a structured, expressive, and unambiguous representation to specify tasks. Specifically, we consider programs, which are composed of control flows (*e.g.* if/else and loops), environmental conditions, as well as corresponding subtasks, as shown in Figure 1. Not only do programs have expressiveness by describing diverse situations (*e.g.* a river exists) and the corresponding subtasks which are required to be executed (*e.g.* mining wood), but they are also unambiguous due to their explicit scoping. To study the effectiveness of using programs as task specifications, we introduce a new problem, where we aim to develop a framework

**Program**

```
def run():
    if is_there[River]:
        mine(Wood)
        build_bridge()
        if agent[Iron]<3:
            mine(Iron)
        place(Iron, 1, 1)
    else:
        goto(4, 2)
    while env[Gold]>0:
        mine(Gold)
```

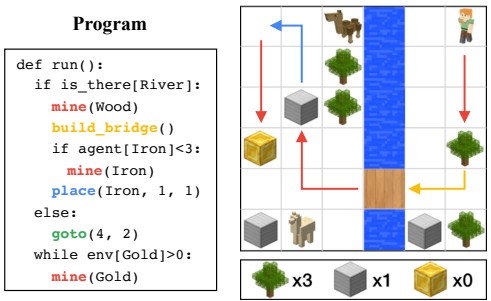

Figure 1: An illustration of the proposed problem. We are interested in learning to fulfill tasks specified by written programs. A program consists of control flows (*e.g.* `if`, `while`), branching conditions (*e.g.* `is_there[River]`), and subtasks (*e.g.* `mine(Wood)`).

Program $p$ := def run() : $s$
Statement $s$ := while$(c)$ : $(s)$ | $b$ | loop$(i)$ : $(s)$
      | if$(c)$ : $(s)$ | elseif$(c)$ : $(s)$ | else : $(s)$
Item $t$ := Gold | Wood | Iron
Terrain $u$ := Bridge | River | Merchant | Wall | Flat
Operators $o$ := $>$ $\geq$ $==$ $<$ $\leq$
Numbers $i$ := A positive integer or zero
Perception $h$ := agent[t] | env[t] | is_there[t] | is_there[u]
Behavior $b$ := mine(t) | goto(i, i)
      | place(t, i, i) | build_bridge() | sell(t)
Conditions $c$ := $h[t]$ $o$ $i$ | $h[u]$ $o$ $i$

Figure 2: The domain-specific language (DSL) for constructing programs. Each program is composed of domain dependent perception, subtasks, and control flows.

which learns to comprehend a task specified by a program as well as perceive and interact with the environment to accomplish the task.

To address this problem, we propose a modular framework, *program guided agent*, which exploits the structural nature of programs to decompose and execute them as well as learn to ground program tokens with the environment. Specifically, our framework consists of three modules: (1) a program interpreter that leverages a grammar provided by the programming language to parse and execute a program, (2) a perception module that learns to respond to conditional queries (*e.g.* `is_there[River]`) produced by the interpreter, and (3) a policy that learns to fulfill a variety of subtasks (*e.g.* `mine(Wood)`) extracted from the program by the interpreter. To effectively instruct the policy with symbolically represented subtasks, we introduce a learned modulation mechanism that leverages a subtask to modulate the encoded state features instead of concatenating them. Our framework (shown in Figure 3) utilizes a *rule-based* program interpreter to deal with programs as well as *learning* perception module and policy when it is necessary to perceive or interact with the environment. With this modularity, our framework can generalize to more complex program-specified tasks without additional learning.

To evaluate the proposed framework, we consider a Minecraft-inspired 2D gridworld environment, where an agent can navigate itself across different terrains and interact with objects, similar to Andreas et al. (2017a); Sohn et al. (2018). A corresponding domain-specific language (DSL) defines the rules of constructing programs for instructing an agent to accomplish certain tasks. Our proposed framework demonstrates superior generalization ability – learning from simpler tasks while generalizing to complex tasks. We also conduct extensive analysis on various end-to-end learning models which learns from not only program instructions but also natural language descriptions. Furthermore, our proposed learned policy modulation mechanism yields a better learning efficiency compared to other commonly used methods that simply concatenate a state and goal.

## 2 RELATED WORK

**Learning from language instructions.** Prior works have investigated leveraging natural languages to specify tasks on a wide range of applications, including navigation (Misra et al., 2018; Tellex et al., 2011; Fried et al., 2018; Vogel & Jurafsky, 2010; Shimizu & Haas, 2009; Branavan et al., 2009; 2012; Misra et al., 2017; Tellex et al., 2011), spatial reasoning for goal reaching (Janner et al., 2018), game playing (Kaplan et al., 2017; Fried et al., 2017; Co-Reyes et al., 2019), and grounding visual concepts (Kaplan et al., 2017; Bahdanau et al., 2019; Andreas et al., 2017b). However, natural language descriptions can often be ambiguous even to humans. Moreover, it is not clear how end-to-end learning agents trained with simpler instructions can generalize well to much more complex ones. In contrast, we propose to utilize a precise and structured representation, programs, to specify tasks.

**Learning from demonstrations.** When a task cannot be easily described in languages (*e.g.* object texture or geometry), expert demonstrations offer an alternative way to provide instructions. Prior works have explored learning from video demonstrations (Finn et al., 2017; Yu et al., 2018b; Xu et al., 2018; Pathak et al., 2018; Stadie et al., 2017; Aytar et al., 2018) or expert trajectories (Duan et al., 2017; Wang et al., 2017b). However, demonstrations can be expensive to obtain and are less expressive about diverging behaviors of a complex task, which are better captured by control flow in programs. Moreover, editing demonstrations such as rearranging the order of subtasks is often difficult.

**Program induction and synthesis.** To learn acquire programmatic skills such as digit addition and string transformations and achieve better generalization, program induction methods (Xu et al., 2018; Devlin et al., 2017a; Neelakantan et al., 2015; Graves et al., 2014; Kaiser & Sutskever, 2016; Reed & De Freitas, 2016; Cai et al., 2017; Xiao et al., 2018) aim to implicitly induce the underlying programs to mimic the behaviors demonstrated in task specifications (*e.g.* input/output pairs, demonstrations). On the other hand, program synthesis methods (Bošnjak et al., 2017; Parisotto et al., 2017; Devlin et al., 2017b; Chen et al., 2019; Shin et al., 2018; Bunel et al., 2018; Liu et al., 2019; Sun et al., 2018; Lin et al., 2018; Liao et al., 2019) explicitly synthesize the underlying programs and execute the programs to perform the tasks. Instead of trying to infer programs from task specifications, we are interested in explicitly executing programs. Also, our framework can potentially be leveraged to obtain program execution results for evaluating program synthesis frameworks when no program executor is available (*e.g.* programs describe real-world activities instead of behaviors in simulation).

**Symbolic planing and programmable agent.** Classical symbolic planning concerns the problem of achieving a goal state from an initial state through a series of symbolically represented executions (Ghallab et al.; Konidaris et al., 2018). Our work shares a similar spirit but assume a task (*i.e.* a program) is given, where the agent is required to learn to ground symbolic concepts (Mao et al., 2019; Han et al., 2019) and follow the control flow. Executing programs with reinforcement learning has been studied in programmable hierarchies of abstract machines (Parr & Russell, 1998; Andre & Russell, 2001; 2002), which provide partial descriptions and subroutines of the desired task. Denil et al. (2017); Lázaro-Gredilla et al. (2018) train agents to execute declarative programs by grounding these well-structured languages in their learning environments. In contrast, our modular framework consists of modules for perceiving the environment and interacting with it by following an imperative program which specifies the task. An extended discussion on the related work can be found in Section C.

## 3 PROBLEM FORMULATION

We are interested in learning to comprehend and execute an instruction specified by a program to fulfill the desired task. In this section, we formally describe our definition of programs, the family of Markov Decision Processes (MDPs), and the problem formulation.

**Program.** The programs considered in this work are defined based on a Domain Specific Language (DSL) as shown in Figure 2. The DSL is composed of perception primitives, action primitives, and control flow. A perception primitive indicates circumstances in the environment (*e.g.* `is_there(River)`, and `agent[Gold]<3`) that can be perceived by an agent, while an action primitive defines a subtask that describes a certain behavior (*e.g.* `mine(Gold)`, and `goto(1,1)`). Control flow includes `if/else` statements, loops, and Boolean/logical operators to compose more sophisticated conditions. A program $p$ is a deterministic function that outputs a desired behavior (*i.e.* subtask) given a history of states $o_t = p(H_j)$, where $H_j = \{s_1, ..., s_t\}$ is a state history with $s \in \mathcal{S}$ denoting a state of the environment, and $o \in \mathcal{O}$ denotes an instructed behavior (subtask). We denote a program as $p \sim \mathcal{P}$, an infinite program set containing all executable programs given a DSL. Note that a discussion on the DSL design principle can be found in Section B.

**MDPs.** We consider a family of finite-horizon discounted MDPs in a shared environment, specified by a tuple $(\mathcal{S}, \mathcal{A}, \mathcal{P}, \mathcal{T}, \mathcal{R}, \rho, \gamma)$, where $\mathcal{S}$ denotes a set of states, $\mathcal{A}$ denotes a set of low-level actions an agent can take, $\mathcal{P}$ denotes a set of programs specifying instructions, $\mathcal{T} : \mathcal{S} \times \mathcal{A} \times \mathcal{S} \to \mathbb{R}$ denotes a transition probability distribution, $\mathcal{R}$ denotes a task-specific reward function, $\rho$ denotes an initial state distribution, and $\gamma$ denotes a discount factor. For a fixed sequence $\{(s_0, a_0), ..., (s_t, a_t)\}$ of states and actions obtained from a rollout of a given policy $\pi$, the performance of the policy is evaluated based on a discounted return $\sum_{t=0}^{T} \gamma^t r_t$, where $T$ is the horizon of the episode.

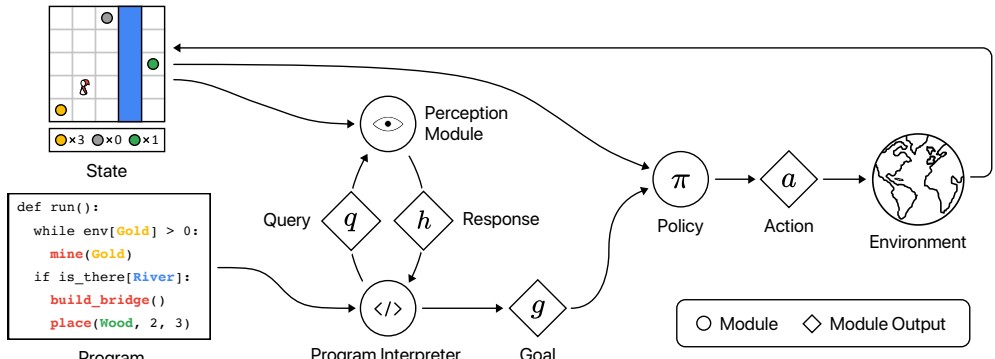

Figure 3: Program Guided Agent. The proposed modular framework comprehends and fulfills a desired task specified by a program. The program interpreter executes the program by altering between querying the perception module with a query $q$ when an environment condition is encountered (*e.g.* env[Gold]>0, is_there[River]) and instructing a policy when it needs to fulfill a goal/subtask $g$ (*e.g.* mine(Gold), build_bridge()). The perception module produces a response $h$ to answer the query, determining which paths in the program should be chosen. The policy takes a sequence of low-level actions $a$ (*e.g.* moveUp, moveLeft, Pickup) interacting with the environment to accomplish the given subtask (*e.g.* mine(Gold)).

**Problem Formulation.** We consider developing a framework which can comprehend and fulfill an instruction specified by a program. Specifically, we consider a sampled MDP with a program describing the desired task. Addressing this task requires the ability to keep track of which parts of the program are finished and which parts are not, perceiving the environment and deciding which paths in the program to take, and performing actions interacting with the environment to fulfill subtasks.

## 4  APPROACH

Accomplishing an instructed task described by a program requires (1) executing the program control flow and conditions, (2) recognizing the situations to infer which path in the program should be chosen, and (3) executing a series of actions interacting with the environment to fulfill the subtasks. Based on this intuition, we design a modular framework with three modules:

- **Program interpreter** (Section 4.1) reads a program and executes it by querying a perception module with environment conditions (*e.g.* env[Gold]>0) and instructing the policy with subtasks (*e.g.* mine(Gold)).
- **Perception module** (Section 4.2) responds to perception queries (*e.g.* env[Gold]>0) by examining the observation and predicting responses (*e.g.* true).
- **Policy (action module)** (Section 4.3) performs low-level actions (*e.g.* moveUp, moveLeft, pickUp) to fulfill the symbolically represented subtasks (*e.g.* mine(Gold)) provided by the program interpreter.

Our key insight is to only *learn* a module when its input or output is associated with the environment (*i.e.* a function approximator is needed) – the perception module learns to ground the queries to its observation and answer them; the policy learns to ground the symbolically represented subtasks and interact with the environment in a trial-and-error way (Section 4.4). On the other hand, we do not learn the program interpreter; instead, we utilize a *rule-based* parser to execute programs An overview of the proposed framework is illustrated in Figure 3.

### 4.1  PROGRAM INTERPRETER

To execute a program instruction, we group program tokens into three main categories: (1) **subtasks** indicate what the agent should perform (*e.g.* mine(Gold)), (2) **perceptions** the essential information extracted from the environment (*e.g.* env[Gold]>0), and (3) **control flows** determine which paths in a program should be taken according to the perceived information (*i.e.* perceptions). Then, we devise a *program interpreter*, which can execute and keep track of the progress by leveraging the

structure of programs. Specifically, it consists of a program line parser and a program executor. The parser first transforms the program into a program tree by representing each line of a program as a tree node. Each node is either a leaf node (subtask) or a non-leaf node (perception or control flow) that has various subroutines as children nodes. The executor then performs a pre-order traversal on the program tree to execute the program, utilizing the parsed contents to alternate between querying the perception module when an environment condition is encountered and instructing the policy when it reaches to a leaf node (subtask). The details and the algorithm are summarized in section A. Note that the *program interpreter* is a rule-based algorithm instead of a learning module.

## 4.2 PERCEPTION MODULE

Determining which paths should be chosen when executing a program requires grounding a symbolically represented query (*e.g.* `is_there[River]` can be represented as a sequence of symbols) and perceiving the environment. To this end, we employ a *perception module* $\Phi$ that learns to map a query and current observation to a response: $h = \Phi(q, s)$, where $q$ denotes a query, and $h$ denotes the corresponding perception output (*e.g.* `true`/`false`). Note that we focus on Boolean perception outputs in this paper, but a more generic perception type can be used (*e.g.* object attributes such as color, shape, and size).

## 4.3 POLICY

When program execution reaches a subtask/leaf node (*e.g.* `mine(Gold)`), the agent is required to take a sequence of low-level actions (*e.g.* `moveUp`, `moveLeft`, `Pickup`) to interact with the environment to fulfill it. To enable the execution, we employ a multitask policy $\pi$ (*i.e.* action module) which is instructed by a symbolic goal (*e.g.* `mine(Gold)`) provided by the program interpreter indicating the details of the corresponding subtask. To learn to perform different subtasks, we train the policy using actor-critic reinforcement learning, which takes a goal vector $g$ and an environment state $s$ and outputs a probabilistic distribution $a$ for low-level actions $a \sim \pi(s_t, g_t|\theta)$. The value estimator used for our policy optimization is also goal-conditioned: $V_\pi(s_t, g_t) = \mathbb{E}[\sum_t \gamma^t R_t|s_0 = s, \pi, g_t]$.

While the most common way to feed a state and goal to a policy parameterized by a neural network is to concatenate them in a raw space or a latent space, we find this less effective when the policy has to learn a diverse set of tasks. Therefore, we propose a modulation mechanism to effectively learn the policy. Specifically, we employ a goal network to encode the goal and compute affine transform parameters $\gamma$ and $\beta$, which are used to modulate state features $e_s$ to $\hat{e}_s = \gamma \cdot e_s + \beta$. Then, the modulated features $\hat{e}_s$ are used to predict action $a$ and value $V$. With the modulation mechanism, the goal network learns to activate state features related to the current goal and deactivate others. An illustration is shown in Figure 4 (a). A more detailed discussion of the related works that utilize similar learned modulation mechanisms can be found in Section D.

## 4.4 LEARNING

To follow a program by perceiving the environment and taking actions to interact with it, we employ two learning modules: a perception module and a policy. In this section, we discuss how each module is trained, their training objectives, and optimization methods. More training details and the architectures can be found in section E.4.1.

### 4.4.1 PERCEPTION MODULE

We formulate training the perception module as a supervised learning task. Given tuples of (query $q$, state $s$, ground truth perception $h_{gt}$), we train a neural network $\Phi$ to predict the perception output $h$ by optimizing the binary cross-entropy loss: $\mathcal{L}_{CE} = -h_{gt}log(h) - (1 - h_{gt})log(1 - h)$. A query such as `is_there[River]` is represented as a sequence of symbols. Note that when perception describes more than a Boolean, training the perception module can be done by optimizing other losses such as categorical cross-entropy loss. We train the perception module only on the queries appearing in the training programs with randomly sampled states, requiring it to generalize to novel queries to perform well in executing testing programs.

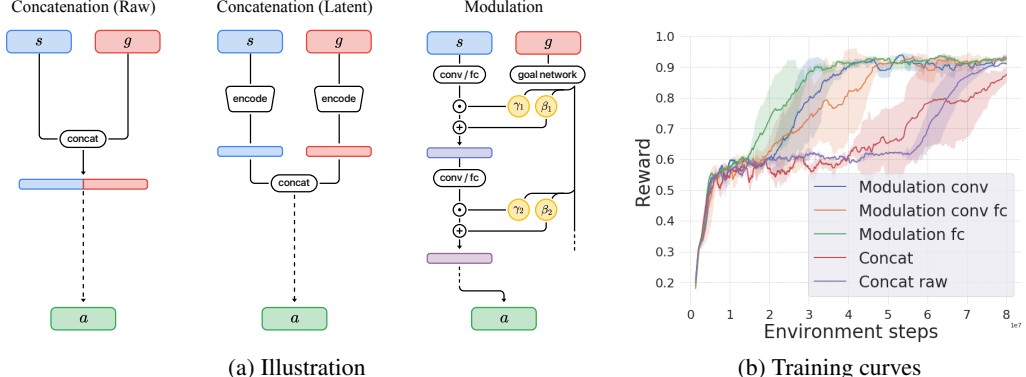

(a) Illustration          (b) Training curves

Figure 4: Learning a multitask policy via learned modulation. (a) A multitask policy takes both a state $s$ and a goal specification $g$ as inputs and produces an action distribution $a \sim \pi(s, g)$. Instead of simply concatenating the state and goal in a raw space or a latent space, we propose to modulate state features $e_s$ using the goal. Specifically, the goal network learns to predict affine transform parameters $\gamma$ and $\beta$ to modulate the state features $\hat{e}_s = \gamma \cdot e_s + \beta$. Then, the final layers use the modulated features to predict actions. (b) We experiment different ways of feeding a state and goal for learning a multitask policy. The training curves demonstrate that all modulation variants, including modulating state feature maps of convolutional layers (*Modulation conv*), modulating state feature vectors of fully-connected layers (*Modulation fc*), or both (*Modulation conv fc*), are more efficient than concatenating the state and the goal in a raw space (*Concat raw*) or a latent space (*Concat*).

### 4.4.2 POLICY

We train the policy using Advantage Actor-Critic (A2C) (Mnih et al., 2016; Dhariwal et al., 2017), which is commonly used for gridworld environments with discrete action spaces. A2C computes policy gradients $A_t \nabla_\theta \log \pi_\theta (a_t | s_t, g_t)$, where $A_t = R_t - V(s_t, g_t)$ is the advantage function based on empirical return $R_t$ starting from $s_t$ and learned value estimator $V(s_t, g_t)$ conditioning on the goal vector $g_t$. We denote the learning rate as $\alpha$, and the policy update rule is as follows:

$$\theta \leftarrow \theta + \alpha \left( A_t \nabla_\theta \log \pi_\theta (a_t | s_t, g_t) + \beta \nabla_\theta H_{\pi_\theta} \right), \tag{1}$$

where $H_{\pi_\theta}$ denotes the policy entropy, where maximizing it improves overall exploration, and $\beta$ determines the strength of the entropy regularization term.

## 5 EXPERIMENTS

Our experiments aim to answer the following questions: (1) Can our proposed framework learn to perform tasks specified by programs? (2) Can our modular framework generalize better to more complex tasks compared to end-to-end learning models? (3) How well can a variety of end-to-end learning models (*e.g.* LSTM, Tree-RNN, Transformer) learn from programs and natural language instructions? (4) Is the proposed learned modulation more efficient to learn a multitask (multi-goal) policy than simply concatenating a state and goal?

### 5.1 EXPERIMENTAL SETUPS

#### 5.1.1 ENVIRONMENT

To evaluate the proposed framework in an environment where an agent can perceive diverse scenarios and interact with the environment to perform various subtasks, we construct a discrete Minecraft-inspired gridworld environment, similar to Andreas et al. (2017a); Sohn et al. (2018). As illustrated in Figure 1, the agent can navigate through a grid world and interact with resources (*e.g.* Wood, Iron, Gold) and obstacles (*e.g.* River, Wall), build tools (*e.g.* Bridge), and sell resources to a merchant visualized as an alpaca. The environment gives a sparse task completion reward of $+1$ when an instruction (*i.e.* an entire program or natural language instruction) is successfully executed. More details can be found in Section E.1.

Table 1: Task completion rate. For each method, we iterate over all the programs in a testing set by randomly sampling ten initial environment states and running three models trained using different random seeds for this method. The averaged task completion rates and their standard deviations are reported. Note that all the end-to-end learning models learning from natural language descriptions and programs suffer from a significant performance drop when evaluated on the more complex testing set.

| Instruction Method | | Natural language descriptions | | Programs | | | | |
| | | Seq-LSTM | Transformer | Seq-LSTM | Tree-RNN | Transformer | Ours (concat) | Ours |
| --- | --- | --- | --- | --- | --- | --- | --- | --- |
| **Dataset** | test | 54.9±1.8% | 52.5±2.6% | 56.7±1.9% | 50.1±1.2% | 49.4±1.6% | 88.6±0.8% | 94.0±0.5% |
| | test-complex | 32.4±4.9% | 38.2±2.6% | 38.8±1.2% | 42.2±2.4% | 40.9±1.5% | 85.2±0.8% | 91.8±0.2% |
| **Generalization gap** | | 40.9% | 27.2% | 31.6% | 15.8% | 17.2% | 3.8% | 2.3% |

### 5.1.2 TASK INSTRUCTIONS

**Programs.** We sample 4,500 programs using our DSL and split them into 4,000 training programs (*train*) and 500 testing programs (*test*). To examine the framework's ability to generalize to more complex instructions, we generate 500 programs which are twice longer and contains more condition branches on average to construct a harder testing set (*test-complex*).

**Natural language instructions.** To obtain the natural language counterparts of those instructions, we asked annotators to construct natural language translations of all the programs. The data collection details, as well as sample programs and their corresponding natural language translations, can be found in Section E.3, and figure 10 respectively. We include a brief discussion on how annotated natural language instructions can be ambiguously interpreted as several valid programs.

### 5.2 TRAINING

During training, we randomly sample programs from the training set as well as randomly sample an environment state to execute the program interpreter. The program interpreter produces a goal to instruct the policy when encountering a subtask in the program. The policy takes actions $a \sim \pi(s, g)$ and receive reward $+1$ only when the entire program is completed. While we do not explicitly introduce a curriculum like Andreas et al. (2017a), this setup naturally induces a curriculum where the policy first learns to solve simpler programs and gain a better understanding of subtasks by obtaining the task completion, which eventually allows the policy to complete more complex programs. Note that the perception module is pre-trained beforehand in a supervised manner. More training details can be found in section E.7.

### 5.3 END-TO-END LEARNING MODELS

In contrast to the proposed modular framework, we experiment with a variety of end-to-end learning models. Considering programs and natural language instructions as sequences of tokens, we investigate two types of state-of-the-art sequence encoders: LSTM (Hochreiter & Schmidhuber, 1997) (*Seq-LSTM*), and Transformers (Vaswani et al., 2017; Devlin et al., 2018) (*Transformer*). To leverage the explicit structure of programs, we also investigate encoding programs using a generalization of RNNs for tree-structured input (Tai et al., 2015; Alon et al., 2019) (*Tree-RNN*). All the models are trained using A2C. The details of these architectures can be found in Section E.4.2.

### 5.4 RESULTS

### 5.4.1 TASK COMPLETION

We train the proposed framework and the end-to-end learning models on training programs and evaluate their performances using the percentage of completed instructions on *test* and *test-complex* sets (shown in Table 1). Our proposed framework achieves a satisfactory *test* performance and only suffers a negligible drop (*i.e.* generalization gap) when it is evaluated on *test-complex* set. This can be attributed to the modular design, which explicitly utilizes the structure and grammar of programs, allowing the two learning modules (*i.e.* perception and policy) to focus on their local jobs. A more detailed failure analysis can be found in Section E.6.

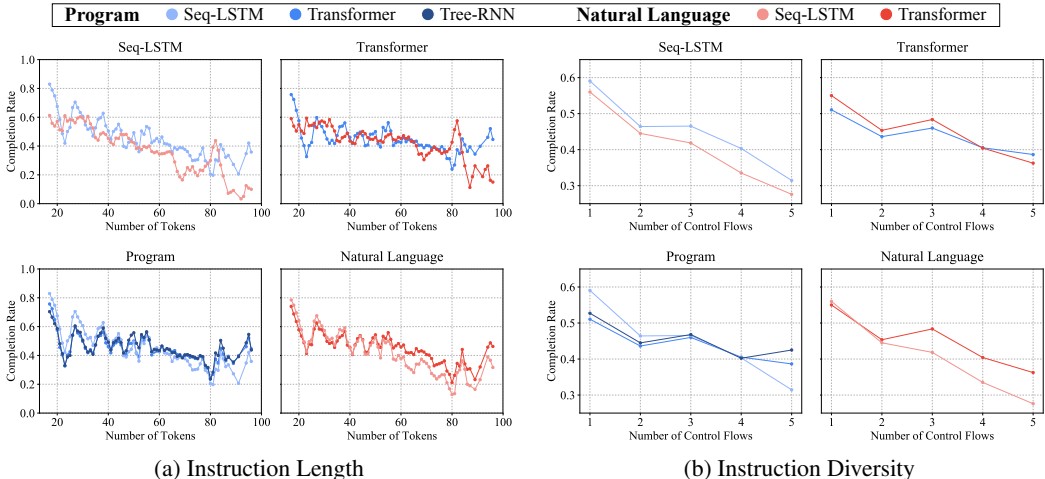

Figure 5: **Analysis on end-to-end learning models:** (a) Models learning from programs generalize better to longer instructions. Transformer is more robust to longer instructions (Upper). Tree-RNN exploiting the program structure generalizes the best, but performs worst for shorter programs (Lower). (b) Seq-LSTM learning from both instructions performs worse as the diversity increases. Transformer learns better from natural language when the instructions are less diverse (Upper). Transformer and Tree-RNN learning from programs are more consistent as the diversity increases, yet perform worse on less diverse instructions (Lower).

On the other hand, all the end-to-end learning models suffer a significant performance drop between *test* and *test-complex* sets, while it is less significant for the models learning from programs, potentially indicating that models learning from instructions with explicit structures can generalize to complex instructions better. Among them, Seq-LSTM achieves the best results on *test* set, but performs the worst on the *test-complex* set. Transformer has smaller generalization gaps, which could be attributed to their multi-head attention mechanism, capturing the instruction semantics better. By leveraging the explicit structure of programs, Tree-RNN achieves the best generalization performance.

### 5.4.2 ANALYSIS

An analysis on the end-to-end learning models with respect to varying instruction length and complexity is shown in Figure 5, where all the instructions from *test* and *test-complex* sets are considered.

**Instruction length.** As shown in Figure 5 (a), both Seq-LSTM and Transformer suffer from a performance drop as the instruction length increases. Seq-LSTM performs better when instructions are shorter, but suffers from generalizing to longer instructions. On the other hand, Transformer may learn on a more semantic level, which leads to similar overall performances across two types of instructions. Tree-RNN leverages the structure of programs and achieves a better performance.

**Instruction diversity.** We define the diversity of a program based on the number of control flows it contains (*i.e.* number of branches). Figure 5 (b) shows a clear trend of performance drop of all the models. Transformer is more robust to diverse instructions which could be attributed to its better ability to learn the semantics. While Seq-LSTM learning from programs are consistently better across different levels of diversities, Tree-RNN demonstrates the most consistent performances.

### 5.5 POLICY MODULATION

We investigate if learning a multitask policy with the learned modulation mechanism is more effective. We compare against the two most commonly used methods: concatenating a state and goal in a raw space (*Concat raw*) or a latent space (*Concat*). An illustration is shown in Figure 4 (a). Since our state contains an environment map, which is encoded by a CNN and MLP, we experiment modulating convoluted feature maps (*Modulation conv*) or feature vectors (*Modulation fc*) or both (*Modulation conv fc*). Figure 4 (b) demonstrate that the proposed policy modulation mechanism is more sample efficient. Table 1 shows that the multitask policy learning using modulation achieves better performance on task completion.

## 6 CONCLUSION

We propose to utilize programs, structured in a formal language, as an expressive and precise way to specify tasks instead of commonly used natural language instructions. We introduce the problem of developing a framework that can comprehend a program as well as perceive and interact with the environment to accomplish the desired task. To address this problem, we devise a modular framework, *program guided agent*, which executes programs with a program interpreter by altering between querying a perception module when a branching condition is encountered and instructing a policy to fulfill subtasks. We employ a policy modulation mechanism to improve the efficiency of learning the multitask policy. The experimental results on a 2D Minecraft environment demonstrate that the proposed framework learns to reliably fulfill program instructions and generalize well to more complex instructions without additional training. We also investigate the performance of various models that learn from programs and natural language descriptions in an end-to-end fashion.

### ACKNOWLEDGMENTS

The authors are grateful for the fruitful discussion with Jiayuan Mao, Yuan-Hong Liao, Youngwoon Lee, Karl Pertsch, and Ayush Jain. The authors would like to thank Sriram Somasundaram for contributing to building the environment and Aleksei Petrenko for the A2C implementation.

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

## A    PROGRAM EXECUTION

We describe our program interpreter in Section 4.1 and provide more details in this section. The program instruction considered in this work contains the following three components: (1) subtasks, (2) perceptions, and (3) control flows. Accordingly, our *Program Interpreter* is designed to consist of (1) a parser to parse each line of the program following the grammar defined by our DSL in figure 2, and (2) a program executor which executes the program conditioning on the parsed contents. The interpreter transforms the program into a tree-like object by exploiting its structure (*i.e.* scopes) and then utilizes the parsed contents to traverse it to execute the program.

A program tree is built by representing each line of a program as a tree node. Each tree node is a data structure containing members: (1) **node.line**, the original line in the program, (2) **node.isLeaf()**, if the current node is a leaf node (*i.e.* subtask), and (3) **node.children**, all the subroutines of the current node (*i.e.* the processes under the scope of the current line of program). The interpreter will parse according to the original line contained in the node, and decide whether to call the policy if it is a leaf node (subtask) or produce a query to call the perception module, deciding which child node (subroutine) to go into.

The subroutines of a node should correspond to proper scoping of the program. For example, in Figure 1 in the main paper, the line `if is_there[River]` has subroutines `mine(Wood)`, `build_bridge()`, `if agent[Iron]<3`, and `place(Iron,1,1)`, but not `mine(Iron)`, which should be `if agent[Iron]<3`'s subroutine.

Once the program tree is built, the program executor will perform a pre-order traversal to initiate the execution. Algorithm 1 summarizes the details of the program execution utilizing the transformed program tree.

---

**Algorithm 1** Program Execution

---

**Require:**  $P$: program to be executed
**Require:**  $s$: environmental state
**Require:**  $\pi$: agent policy parameterized by $\theta$, $\Phi$: perception module
**Require:**  node: has member node.line as the original program line and children nodes node.children
 1:  **procedure** EXECUTE(node)
 2:      **if** node.isLeaf() **then**
 3:          subtask = parse_subtask(node.line)
 4:          $\pi$(subtask, $\theta$)                                        ▷ Calls the agent policy to execute the subtask
 5:      **else**
 6:          control_flow, perception_query = parse_ctrl_percept(node.line)
 7:          $h = \Phi$(perception_query, $s$)          ▷ Calls the perception module with a query and state
 8:          **control_flow** $h$                  ▷ *e.g.* if, while, loop, calls the subroutines depending on $h$
 9:              **for** child in node.children **do**
10:                  EXECUTE(child)
11:              **end for**
12:      **end if**
13: **end procedure**

---

## B    DSL DESIGN PRINCIPLE

Since different domains require different DSLs, we aim to design our DSL by following a design principle that would potentially allow us to easily adapt our DSL to different domain. Specifically, we develop a DSL design principle that considers a general setting where an agent can perceive and interact with the environment to fulfill some tasks. Accordingly, our DSL consist of control flows, perceptions, and actions. While control flows are domain independent, perceptions and actions can be designed based on the domain of interest, which would require certain expertise and domain knowledge. We aim to design our DSL that is (1) intuitive: the actions and perceptions are intuitively align with human common sense, (2) modular: actions are reasonably distinct and can be used to compose more complex behaviors, and (3) hierarchical: a proper level of abstraction that enables describing long-horizon tasks.

## C    EXTENDED RELATED WORK

We present an extended discussion of the related work in this section.

**Multitask reinforcement learning.** To achieve multi-task reinforcement learning, previous works devised hierarchical approaches where an RL agent is trained to achieve a series of subtasks to accomplish the task. In Andreas et al. (2017a), a sequence of policy sketches is predefined to guide an agent towards the desired goal by leveraging modularized neural network policies. Oh et al. (2017) propose to learn a controller to predict to either proceed, revert, or stay at a current subgoal, which is sampled from a list of simple symbolic instructions. In this paper, hierarchical tasks are described by programs with increased diversity through branching conditions, and therefore our framework is required to determine which branches in a program should be executed. On the other hand, the framework proposed by Sohn et al. (2018) requires a subtask graph describing a set of subtasks and their dependencies and aims to find the optimal subtask to execute. This is different from our problem formulation where the agent is asked to follow a given program/procedure.

**Hierarchical reinforcement learning.** Our work is also closely related to hierarchical reinforcement learning, where a meta-controller learns to predict which sub-policy to take at each time step (Kulkarni et al., 2016; Bacon et al., 2017; Dilokthanakul et al., 2017; Frans et al., 2018; Vezhnevets et al., 2017; Lee et al., 2019; Bakker et al., 2004; Nachum et al., 2018; Mao et al., 2018). Previous works also investigated in explicitly specifying sub-policy with symbolic representations for meta-controller to utilize, or an explicit selection process of lower-level motor skills (Mülling et al., 2013; Tianmin Shu, 2018).

**Programmable agents.** We would like to emphasize that our work differs from programmable agents (Denil et al., 2017) in motivation, problem formulations, proposed methods, and contributions. First, Denil et al. (2017) concern declarative programs which specify what to be computed (e.g. a target object in a reaching task). However, the programs considered in our work are imperative, which how this is to be computed (i.e. a procedure). Also, Denil et al. (2017) consider only one-liner programs that contain only AND, OR, and object attributes. On the other hand, we consider programs that are much longer and describe more complex procedures. While Denil et al. (2017) aim to generalize to novel combinations of object attributes, our work is mainly interested in generalizing to more complex tasks (i.e. programs) by leveraging the structure of programs.

**Programs vs. natural language instructions.** In this work, we advocate utilizing programs as a task representation and propose a modular framework that can leverage the structure of programs to address this problem. Yet, natural language instructions enjoy better accessibility and are more intuitive to users who do not have experience in programming languages. While addressing the accessibility of programs or converting a natural language instruction to a more structural form is beyond the scope of this work, we look forward to future research that leverages the strengths of both programs and natural language instructions by bridging the gap between these two representations, such as synthesizing programs from natural language (Lin et al., 2018; Desai et al., 2016; Raza et al., 2015), semantic parsing that bridges unstructured languages and structural formal languages (Yu et al., 2018a; Yin & Neubig, 2018), and naturalizing program (Wang et al., 2017a).

## D    DISCUSSIONS ON LEARNED MODULATION MECHANISMS

To fuse the information from an input domain (*e.g.* an image) with another condition domain (*e.g.* a language query, image such as segmentation map, noise, etc.), a wide range of works have demonstrated the effectiveness of predicting affine transforms based on the condition to scale and bias the input in visual question answering (Perez et al., 2018; 2017), image synthesis (Almahairi et al., 2018; Karras et al., 2019; Park et al., 2019; Huh et al., 2019), style transfer (Dumoulin et al., 2017), recognition (Hu et al., 2018; Xie et al., 2018), reading comprehension (Dhingra et al., 2017), few-shot learning (Oreshkin et al., 2018; Lee & Choi, 2018), etc. Many of those works present an extensive ablation study to compare the learned modulation against traditional ways to merge the information from the input and condition domains.

Recently, a few works have employed a similar learned modulation technique to reinforcement learning frameworks on learning to follow language instruction (Bahdanau et al., 2019) and meta-reinforcement learning (Vuorio et al., 2018; 2019). However, there has not been a comprehensive

ablation study that suggests fusing the information from the input domain (*e.g.* a state) and the condition domain (*e.g.* a goal or a task embedding) for the reinforcement learning setting. In this work, we conduct an ablation study in our 2D Minecraft environment where an agent is required to fulfill a navigational task specified by a program and show the effectiveness of learning to modulate input features with symbolically represented goal as well as present a number of modulation variations (*i.e.* modulating the fully-connected layers or the convolutional layers or both). We look forward to future research that verifies if this learned modulation mechanism is effective in dealing with more complex domains such as robot manipulation or locomotion.

# E    ADDITIONAL EXPERIMENTAL DETAILS

## E.1    ENVIRONMENT DETAILS

In the following paragraphs, we provide some details of the environment used in this work.

**Objects in the environment.** The major environmental resources that the agent can interact with are: wood, gold, and iron. There is a certain probability that the environment will contain a river, which the agent cannot go across unless a bridge is built (or pre-built). The environment is surrounded by brick walls, which draws the boundaries of the world.

**Agent action space.** The agent's actions are (1) **crafting actions**: including mining (collecting resources), placing, building a bridge, and selling an item; and (2) **motor actions**: including moving to four directions (up, down, left, right). The crafting actions are only allowed on the current grid cell the agent is standing on, *e.g.* the subtask mine(gold) requires the agent to navigate to a specific location containing a gold with motor actions, and then perform the crafting action mine at the current location. To build a bridge, the agent should consume one of the wood it possesses. To sell an item, the agent needs to travel to a merchant. With certain probabilities, there can be 2 to 4 merchants at different locations.

**Initialization.** During training, when each training program is sampled, a valid environment will be randomly initialized, where validity refers to the property that the agent will be able to successfully follow the program with sufficiently provided environmental resources. At test time, we pre-sample 20 valid initialization of the environment with 20 different random seeds to ensure the validity of the two test sets.

**Agent observation space (state representations).** The state used in our reinforcement learning policy consists of an environment map $s_{\text{map}}$ and an inventory status of the agent $s_{\text{inv}}$. $s_{\text{map}}$ is of size $10 \times 10 \times 9$, where each channel-wise slice of size $10 \times 10 \times 1$ represents the binary existence of certain objects at a specific location, *e.g.* if $(3, 4, 2)$ is 1, it means there is a gold at location $(3, 4)$ (environment objects in channel dimension are zero-indexed). The objects represented by the channels are ordered as follows: wood, iron, gold, agent, wall, goal (2-D representation of intended goal coordinates), river, bridge, and merchant. The agent inventory status $s_{\text{inv}}$ is augmented with the agent location, resulting in a 1-d integer vector of size 5. The ordered entry of such vector is as follows: agent's wood counts, agent's iron counts, agent's gold counts, agent's location coordinate x, and then y.

**Goal representations.** For our proposed framework, we represent the goal of the subtask as an 1-D vector of size 10, produced by the program interpreter. The first five entries of the goal vector is a one-hot representation of the subtask: goto, place, mine, sell, build_bridge. The 6th to 8th entries are one-hot representation of the resources: wood, iron, and gold. The last two entries are the intended goal locations. For example, place(iron,3,5) will be represented as $[0, 1, 0, 0, 0, 0, 1, 0, 3, 5]$. For end-to-end learning models, such goal representation is produced by the input encoder as a continuous latent vector representation.

**Exemplar environment maps.** We show several exemplary rendered environment maps in Figure 6. As can be seen, the essential resources such as wood, gold, iron are represented as block objects, where the merchant is depicted by an alpaca. The agent is shown as a female human character. River grids, with bridge blocks built on it is shown as the blue grid cells, where the bridge which should be transformed by wood is of wooden texture. The Boundaries can be seen in the surroundings represented as brick wall grids.

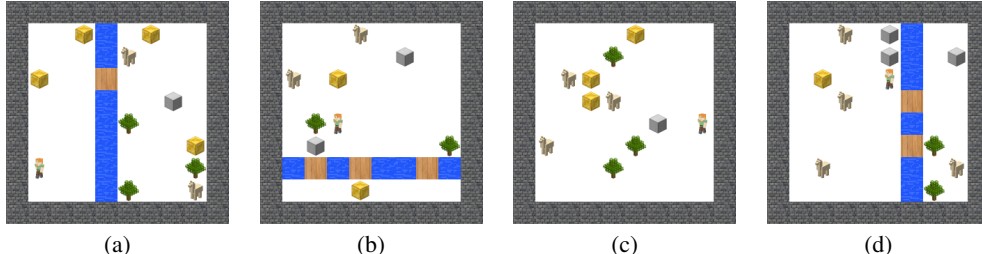

|        |        |        |        |
| :----: | :----: | :----: | :----: |
| (a)    | (b)    | (c)    | (d)    |

Figure 6: Exemplar rendered environment map. The agent, objects, and stuff are represented as blocks with their corresponding textures. Specifically, the `agent` is represented as a female character. `gold` is represented as a golden block, `wood` is shown as a tree, and `iron` is represented as a sliver block. `River` is shown as a blue grid with water texture while `bridge` is presented as wooden grid. `merchant` is shown as an alpaca, which is supposed to transport the sold objects. Notice that there are 2 `merchant`s in (a) and (b), while (c) and (d) contains 3 and 4 of them, respectively. The boundaries of the map are shown as brick walls.

## E.2  GROUND TRUTH PERCEPTIONS FOR END-TO-END LEARNING BASELINES

Since we train our perception module using ground truth information, we provide the ground truth perception information to all our baselines yet failed to elaborate this in the original paper. Specifically, at every time step, we feed the ground truth perception (*i.e.* the answer to the queries such as `env[Gold]>0` and `is_there[River]`) to the baseline models. The ground truth perception is represented as a vector that has a dimension of the number of all possible queries, and each element corresponds to a binary answer to a query. Therefore, the baseline models can learn to utilize this ground truth information to infer the desired subtasks. During testing time, the baseline models can still access to all this ground truth perception information, even though it is usually not possible in practice. On the contrary, during testing time, our perception module predicts the answer to given queries and the performance of the whole framework depends on the predicted answers.

## E.3  TASK INSTRUCTIONS DETAILS

**Programs.** We generate the program sets by sampling program tokens with normal distributions, and constructing them according to the DSL grammar we define. The training set is composed of on average 32 tokens and 4.6 lines; the more complex test set, *i.e. test-complex*, contains on average 65 tokens and 9.8 lines. We include the plotted statistics of various essential properties for the three datasets in Figure 11, Figure 12, and Figure 13, respectively. Note that the maximum indent of a program is the maximum depth of its scope or the height of its transformed program tree. The number of recurring procedures includes both `while` and `loop`.

**Natural language instructions.** For each of the three program sets, we chunked them into several subsets of programs and assign them to annotators for their corresponding natural language translations. The annotators were instructed to read the provided DSL to understand the details of program syntax as well as some exemplary translations before they are allowed to start the task. The annotators were encouraged to give diverse and colloquial translations to avoid constantly giving dull line-by-line translations. The collected (translated) natural language instructions were then cleansed with spell checks and grammatical errors fixes. On average, the annotators used 27, 28, and 61 words to describe the instructions for the train, test, and test-complex sets respectively. The total vocabulary size of the natural language instructions is of 448.

**Qualitative results on natural language analysis.** We show several example data points from our testing sets in Figure 10. The leftmost column displays natural language instructions, the middle column shows our sampled ground truth programs, while the rightmost column illustrates how language can be ambiguous and lead to possible alternative interpreted programs.

## E.4  NETWORK ARCHITECTURES

The proposed framework and the end-to-end learning baselines are implemented in TensorFlow (Abadi et al., 2016).

### E.4.1 OUR FRAMEWORK

**Perception module.** The perception module takes a query $q$ and a state $s$ as input and outputs a response $h$. A query has a size of $6 \times 186$, since the longest query has a length of 6 and 186 is the dimension of one-hot program tokens. Shorter queries are zero-padded to this size.

The state map $s_{\text{map}}$ is encoded by a CNN with four layers with channel size of 32, 64, 96, and 128. Each convolutional layer has kernel size 3 and stride 2 and is followed by ReLU nonlinearity. The final feature map is flattened to a feature vector, denoted as $f_m$.

The state inventory $s_{\text{inv}}$ is encoded by a two-layer MLP with a channel size of 32 for both layers. Each fully-connected layer is followed by ReLU nonlinearity. The resulting feature vector is denoted as $f_i$.

Each token in the query is first encoded by a two-layer MLP with a channel size of 32 for both layers. Each fully-connected layer is followed by ReLU nonlinearity. Then, all the query token features are concatenated along the feature dimension to a single vector. This vector is then encoded by another two-layer MLP with a channel size of 32 for both layers. Each fully-connected layer is followed by ReLU nonlinearity. The resulting feature vector is denoted as $f_q$.

All encoded features ($f_m$, $f_i$, and $f_q$) are then concatenated along the feature dimensions to a single vector. This vector is processed by a three-layer MLP with a channel size of 128, 64, and 32. Each fully-connected layer is followed by ReLU nonlinearity. Finally, a linear fully-connected layer produces an output with a size of 1, which should have a higher value if the response of the query is `true` and lower otherwise.

**Policy.** The policy takes a goal $g$ and a state $s$ as input and outputs an action distribution $a$, where the state is encoded by two types of modules: (1) a four-layer CNN encoder to encode the state map $s_{\text{map}}$, and (2) a two-layer MLP to encode the agent inventory status $s_{\text{inv}}$.

The goal $g$ is encoded by a two-layer MLP with a channel size of 64 for both layers. Each fully-connected layer is followed by ReLU nonlinearity. The resulting feature vector is denoted as $f_g$. Given the encoded goal vector, we employ four linear fully-connected layers to predict modulation parameters $\{\gamma_i, \beta_i\}_{\{1,\ldots,4\}}$ for the state CNN encoder, where $\gamma_1$ and $\beta_1$ have size 32, $\gamma_2$ and $\beta_2$ have size 64, $\gamma_3$ and $\beta_3$ have size 96, and $\gamma_4$ and $\beta_4$ have size 128. Note that these modulation parameters are predicted for modulating convolutional features (*i.e. modulation conv*). For *modulation fc*, a linear fully-connected layer is used to produce $\gamma^{fc}$ and $\beta^{fc}$ with size 64.

A state map $s_{\text{map}}$ is encoded by four-layer CNN with channel size of 32, 64, 96, and 128. Each convolutional layer has kernel size 3, strides 2, and is followed by ReLU nonlinearity. After each convolutional layer, the produced feature maps $e$ are modulated to $\gamma \cdot e + \beta$, where $\gamma$ and $\beta$ are broadcast along spatial dimensions. The final feature map is flattened to a feature vector and denoted as $f_m^\pi$.

A state inventory $s_{\text{inv}}$ is encoded by a two-layer MLP with channel size of 64 for both layers. Each fully-connected layer is followed by ReLU nonlinearity. The resulting feature vector is denoted as $f_i^\pi$.

The two encoded features ($f_m^\pi$ and $f_i^\pi$) are then concatenated along the feature dimension. Two fully-connected layers are used to process the feature with a channel size of 64 for both layers. Each layer is followed by ReLU nonlinearity. The final encoded feature $u$ is then modulated to $\gamma^{fc} \cdot u + \beta^{fc}$ if *modulation fc* is used.

Finally, the modulated features $\hat{u}$ are used to produce an action distribution $a$ and a predicted value $V$ using two separated MLPs. Each MLP has two fully-connected layers with a channel size of 64 for both layers. A linear layer then outputs a vector with a size of 8 (the number of low-level actions). Another linear layer outputs a vector with a size of 1 as the predicted value.

### E.4.2 END-TO-END LEARNING MODELS

In addition to the input encoder, the end-to-end learning models can utilize a mechanism to remember what subtasks from the instructions have been accomplished. The agent can then explicitly memorize where it stands in the instruction while completing the task. We augment such memorization mechanism utilizing the memory of another LSTM network, taking as inputs the encoded states

throughout the execution trajectory. After agent taking each action, the last hidden state encoding the trajectory up to the current step is used to compute attention scores to pool the outputs of the input encoders. For Tree-RNN encoder, we simply concatenate the hidden representation from memorization LSTM with the root representation of Tree-RNN before feeding them to subsequent layers. The agent policy network then learns to perform task conditioning on this attention-pooled latent instruction vector.

We provide details of our various end-to-end learning models in Table 2. Program token embedding is jointly trained with learning the whole module, while GloVe Pennington et al. (2014) (50-D version) is used for word embedding when instructions are natural languages.

| Model | Parameters | Details |
|---|---|---|
| Seq-LSTM | 0.62M | LSTM size of 128, both program and word embeddings are of dimension 50. Attention LSTM size of 128. Attention weights of size $[256 \times 128]$, with bias of size $[128]$. Word embeddings utilize pre-trained GloVe. |
| Tree-RNN | 0.51M | Program embeddings are of dimension 128. Attention LSTM size of 128. Composition module (to aggregate all the children representation of a node) is of size $[128 \times 128]$, and output projection weights of size $[128 \times 128]$, with bias of size $[128]$. The program embeddings are average pooled across the same program line, so that each line will be mapped to a fixed dimension representation. The composition layer is applied when combining pooled embedding from all the children of a node. |
| Transformer | 2.63M | Number of hidden layers: 2, with 8 attention heads, and intermediate size of 256. Hidden size is 128. No dropout is applied. |

Table 2: Architectural details for end-to-end learning models

### E.5 RAW RGB INPUT

To verify if our framework can be extended to using high-dimensional raw state inputs (*i.e.* RGB image) as inputs where a hand-crafted policy or perception module might not be easy to obtain, we performed an additional experiment where the perception module and the policy are trained on raw RGB inputs instead of the symbolic state representation. The results suggest that our framework can utilize RGB inputs while maintaining similar performance (93.2% on the test set) and generalization ability (91.4% on the test-complex set).

### E.6 FAILURE ANALYSIS

To gain a better understanding of how our proposed framework and the end-to-end learning models work or fail, we conduct detailed failure analysis on the execution traces of our model. The analysis is organized as follows:

- We first present an analysis of our framework on the subtasks that appear to be the first failed subtask, which immediately leads to failing the whole task. This analysis sheds some light on which subtasks most commonly cause the failure of task execution. (Section E.6.1)
- We show an analysis of how many time steps each successfully executed subtask takes on average for our framework, through which we explain which subtasks we find to be harder than others. (Section E.6.2)
- We show additional visualizations on the completion rates of different end-to-end learning models plotted with metrics not shown in the main paper, where we aim to deliver a more complete view of how these models perform. (Section E.6.3)

### E.6.1 FIRST FAILURE RATE OF SUBTASKS

As the first step of failure analysis, we want to get an idea of which subtasks cause the failure of the model in executions more often. To make this possible, we define "first failed subtask" as the first subtask that ends as a failure in an unsuccessful execution of a program. Based on this definition, we

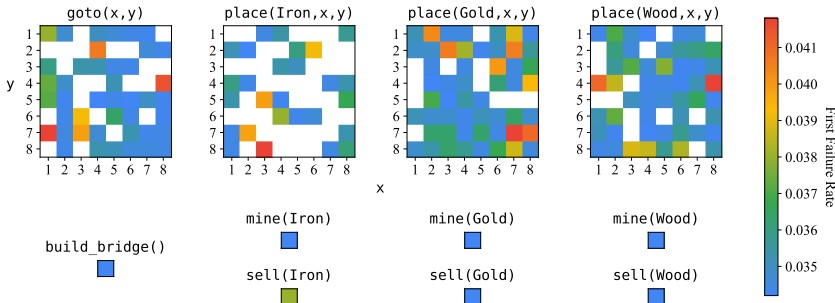

Figure 7: **First failure rate of subtasks.** Every colored grid shows the first failure rate of each subtask. From top-left to bottom-right, each block of grids show the results for subtask category `goto`, `place`, `build_bridge`, `mine`, and `sell`. Warmer colors indicate higher first failure rate; while colder colors indicate lower first failure rate. White grids indicate subtasks that either never occurs as first failed task in any execution or do not exist in the executions that lead to this figure.

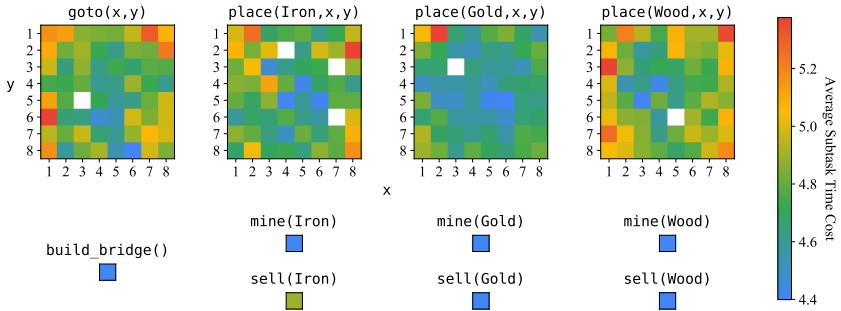

Figure 8: **Average time cost of subtasks.** The setup of this plot is similar to that of Figure 7. Warmer colors indicate higher average subtask time cost; while colder colors indicate lower average subtask time cost. White grids indicate subtasks that do not exist in the executions that lead to this figure.

further define "first failure rate" as the percentage that an occurrence of a specific subtask turns out to be the first failed subtask of the execution that includes it.

We collect the first failure rate of all subtasks for the result we obtain from running our full model over the more complex test set, *i.e. test-complex*. The results are plotted in a visually interpretable format in Figure 7. As seen in the figure, subtasks in `goto` and `place` categories are more likely to be the first failed subtask than subtasks in `build_bridge`, `mine`, and `sell` categories. Within the `goto` and `place` subtask categories, subtasks requiring the agent to navigate to grid cells nearby the border of the world has a higher first failure rate than ones nearby the center of the world. This shows that these tasks mentioned above are more prone to failure than other subtasks.

### E.6.2 AVERAGE TIME COST OF SUBTASKS

Continuing from the previous analysis, we show the average time cost of all successful subtask executions in Figure 8. As can be seen in the figure, subtasks in `build_bridge`, `mine`, and `sell` categories take relatively smaller number of time steps to complete. In `goto` and `place` categories, the closer to border the subtask requires the agent to reach, the more time consuming it gets for the agent to complete the subtask. This corresponds to the finding in the analysis of first failure rates that subtasks with destinations close to the border are more likely to fail. In other words, the closer to border the agent has to reach, the more likely it is to fail the subtask.

### E.6.3 ADDITIONAL ANALYSIS ON END-TO-END LEARNING MODELS COMPLETION RATES

To conclude failure analysis, we focus on the variation of completion rates of program executions with respect to different conditioning variables. As shown in Figure 9, plots showing the trends of completion rates while evaluating with different independent variables show that execution failure is

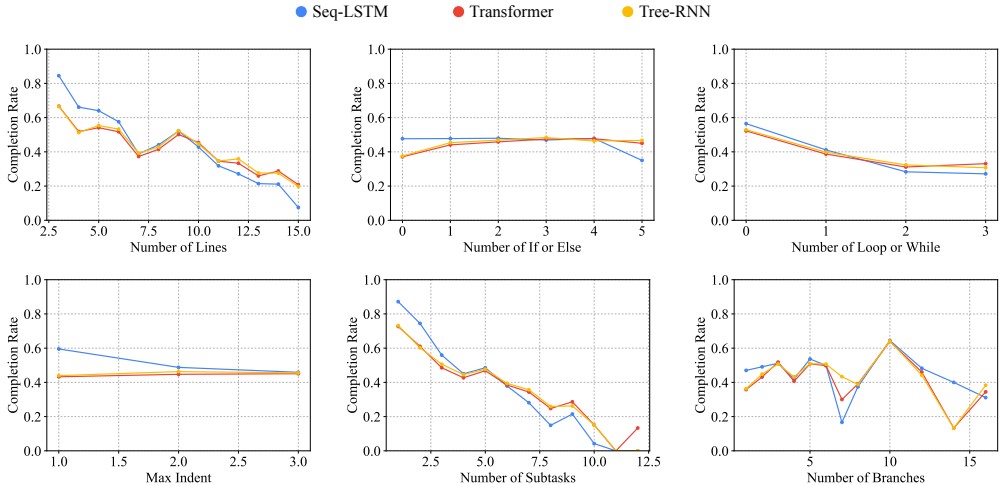

Figure 9: Additional analysis on completion rates. The results of executing program instructions on both datasets are used to produce the six plots above. In each plot, each color corresponds to a different model that we propose. For the two rightmost plots, there is a very small number of outliers that extends out of the right boundary of the plot that we omit for visual interpretability reasons. Please note that the use of colors in this figure is not the same as that in Figure 5 of the main paper.

more common in cases when the program consists of larger number of lines, more loops and while statements, and larger number of subtasks. Note that this subtask count is only a summation of the occurrence counts of each subtask in a program, which does not accurately reflect the number of executions each subtask is being invoked (*i.e.* it does not reflect the repetitive counts when there is a loop).

Meanwhile, the effect of the number of if and else statements and the maximum indent values of programs on completion rates seem to vary across different models. For Seq-LSTM model, having a larger number of if and else statements or having a larger max indent value results in more failures; while for Transformer and Tree-RNN models, having larger values above results instead in fewer failures. This is probably since Transformer and Tree-RNN models are designed in a way that deals with hierarchical structures with jumps in instruction executions better (this point is also mentioned in the main paper). Despite this difference in effects, the overall change in performance when the number of if and else statements and the maximum indent value change is much less significant than that in the previous case.

During our analysis, we also designed an algorithm to calculate an estimate to the number of branches a program has. Here, the number of branches is defined by the number of distinct sets of lines that a program can be executed. For a program without control flows (no if, else-if, else, loop, and while statements), the number of branches is always 1. For if, else-if, and else statements, the exact number of branches these statements incur can be calculated easily. In cases of the loop and while statements, we treat loops as being executed only once and while statements as if statements when we calculate the number of branches. The result shown in the analysis does not reveal a clear trend. We attribute this result to two possibilities – either the metric we create is not accurate enough, or it is not a very suitable metric to be inspected.

### E.7 HYPERPARAMETERS

We use the following hyperparameters to train A2C agents for our model and all the end-to-end learning models: learning rate: $1 \times 10^{-3}$, number of environment: $64$, number of workers: $64$, and number of update roll-out steps: $5$.

### E.8 COMPUTATIONAL RESOURCES

We train all our models on a single Nvidia Titan-X GPU, in a 40 core Ubuntu 16.04 Linux server.

| # | Language Instructions | Ground Truth Program | Alternative Interpretation |
|---|---|---|---|
| (a) | If there is a river, build a bridge. Repeat the followings 3 times: mine a gold, and if environment has no more than 8 gold, mine iron, and then sell an iron. | ```def run():    if is_there[River]:        build_bridge()    loop(3):        mine(Gold)        if env[Gold] <= 8:            mine(Gold)    sell(Iron)``` | ```def run():    if is_there[River]:        build_bridge()    loop(3):        mine(Gold)    if env[Gold] <= 8:        mine(Gold)    sell(Iron)``` |
| (b) | Place an iron on (7,2) and repeat 4 times, if agent has no more than 9 iron then sell a gold. | ```def run():    place(Iron, 7, 2)    loop(4):        if agent[Iron] <= 9:            sell(Gold)``` | ```def run():    loop(5):        place(Iron, 7, 2)    if agent[Iron] <= 9:        sell(Gold)``` |
| (c) | Mine wood first. If agent has more than 3 iron, mine wood. If there is gold in the environment, place iron at (3,7). | ```def run():    mine(Wood)    if agent[Iron] >= 4:        mine(Wood)        if is_there[Gold]:            place(Iron, 3, 7)``` | ```def run():    mine(Wood)    if agent[Iron] >= 4:        mine(Wood)    if is_there[Gold]:        place(Iron, 3, 7)``` |

Figure 10: Exemplar data and languages ambiguity. The goal of the examples above is to show that natural language instructions while being flexible enough to capture the high-level semantics of the task, can be ambiguous in different ways and thus might lead to impaired performance. In example (a), the modifier "repeat the following 3 times" has an unclear scope, resulting in two possible interpretations shown in program format on the right side; in example (b), "repeat 4 times" can be used to modify either the previous part of the description or the latter part of it, resulting in ambiguity; in example (c), the last sentence starting with "If" has unclear scope. In all of the above cases, a model that learns to execute instructions presented in natural language format might fail to execute the instructions successfully because of the ambiguity of the language instructions.

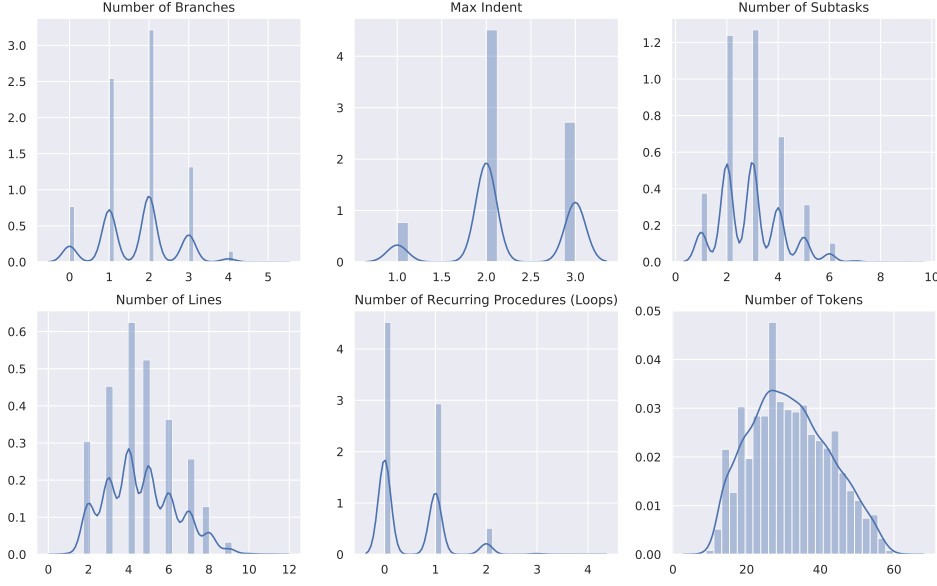

Figure 11: Program set statistics for training set (*train*).

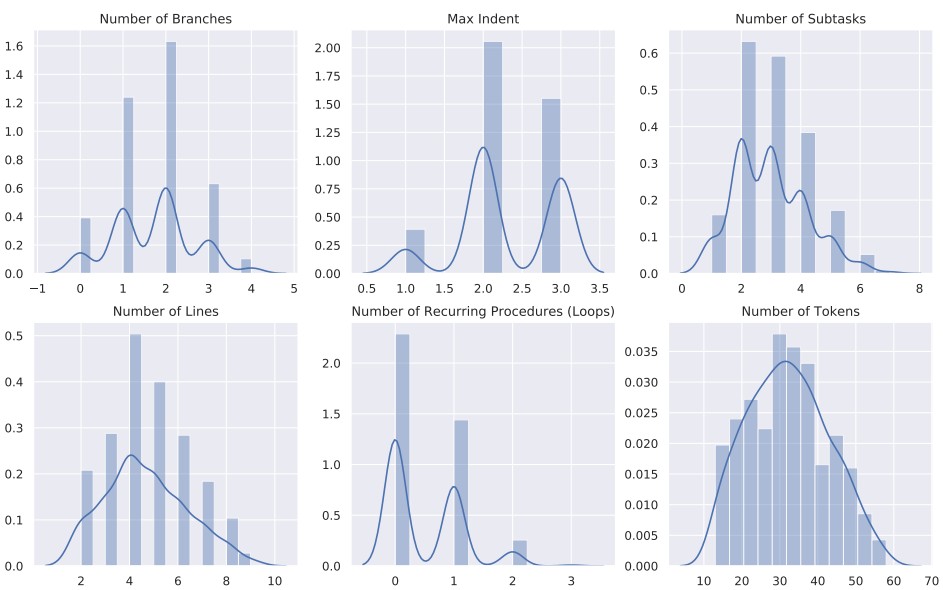

Figure 12: Program set statistics for same complexity testing set (*test*).

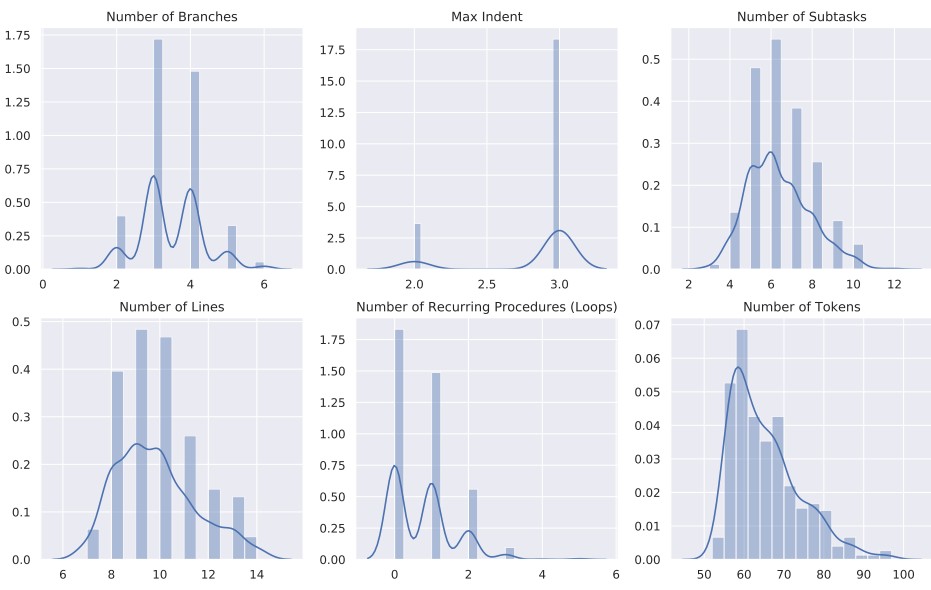

Figure 13: Program set statistics for more complex testing set (*test-complex*).

