# OpenReview forum: "Program Guided Agent"
_ICLR.cc/2020/Conference — Accept (Spotlight)_

### Official Review · AnonReviewer1 · 2019-10-23
**Official Blind Review #1**

**Rating:** 8

**Review:**

Update: I thank the reviewers for their extensive rebuttal and revision of the paper addressing all of my concerns. I have increased my score.

Summary
This paper investigates an important direction: How can RL agents make use of high-level instructions and task decompositions formalized as programs? The authors propose a model for a program guided agent that, conditioned on a program, interprets the program, executes it to query a perception module and subsequently proposes subgoals to a low-level action module. The method outperforms LSTM and Transformer baselines on a Minecraft-like task and generalizes to programs larger than the one seen during training.

Strengths
Contribution in the important direction of training RL agents with instructions and prior knowledge, here in the form of programs
Clearly written paper with good illustrations of the model
Good performance on generalization task of acting in environments where the programmatic instructions are longer than those seen during training

Weaknesses
One of the contributions of the paper is a modulation mechanism (Section 4.3) on the state features that incorporates a goal-conditioned policy. However, a very related approach has been proposed by Bahdanau, Dzmitry, et al. "Learning to Understand Goal Specifications by Modelling Reward." ICLR 2019. They introduced FILM layers that modulate the layers in a ConvNet conditioned on a goal representation. This should be discussed and compared to in the paper.
I am surprised there is no comparison to other work that conditions on programs or hierarchical RL approaches. For example, the authors mention various works in Section 2, but fail to compare to them or at least explain why a comparison would not be possible.
Another point of criticism is that the authors do not use an existing environment, but instead a Minecraft-inspired one similar to Andreas et al, Oh et al. and Sohn et al. This makes a comparison to prior work hard and I would like to understand in what way previous environments were inadequate for the research carried out here.
One aspect that I found most interesting in this paper is that the authors also let annotators map the given programs into natural language form. However, there is no discussion of these results. Similarly, there are interesting qualitative analyses in the appendix of the paper that I only stumbled upon by chance. I believe these should be referenced and a short summary should be integrated into the main part of the paper. I would particularly like to see a discussion of limitations already in the main part of the paper.

Minor Comments
p1: I like the motivation of cooking recipes for work on program conditioned learning. There is in fact a paper (probably multiple) from the NLP community that I think could be cited here. The one that comes to my mind is: Malmaud, Jonathan, et al. "Cooking with semantics." Proceedings of the ACL 2014 Workshop on Semantic Parsing. 2014.
p1: I agree with the argument that programs might be favored over natural language to specify goals as they are unambiguous. However, I think this can also be seen as a drawback. Natural language allows us to very efficiently share information, maybe sometimes information that is only disambiguated through observations in the environment. Another advantage is that natural language for instructing learning agents (like people) is abundant on the web, while programs are not.
p2: "that leverages grammar" -> "that leverages a grammar"
p2: "we propose to utilize an precise" -> "we propose to utilize a precise"
p2: For learning from video demonstrations, an important prior work is Aytar, Yusuf, et al. "Playing hard exploration games by watching youtube." Advances in Neural Information Processing Systems. 2018.
p3: A deep learning program synthesis work prior to the ones mentioned here is Bošnjak, Matko, et al. "Programming with a differentiable forth interpreter." Proceedings of the 34th International Conference on Machine Learning-Volume 70. JMLR. org, 2017.
p5: Would it make sense to also compare to a purely hand-crafted programmatic policy? I am missing a justification why learning is strictly necessary in the environment considered in this work.
p6 Section 4.4.1: I believe the explanation of the perception module would benefit from a concrete example.
Questions to Authors

**Experience Assessment:**

I have published one or two papers in this area.

**Review Assessment: Checking Correctness Of Derivations And Theory:**

I assessed the sensibility of the derivations and theory.

**Review Assessment: Checking Correctness Of Experiments:**

I assessed the sensibility of the experiments.

**Review Assessment: Thoroughness In Paper Reading:**

I read the paper at least twice and used my best judgement in assessing the paper.

---

> ### Author Response · Authors · 2019-11-14
> **Response to Reviewer #1 (3/3)**
>
>
> Q: Would it make sense to also compare to a purely hand-crafted programmatic policy? I am missing a justification why learning is strictly necessary in the environment considered in this work.
>
> A: We aim to develop a program guided agent that perceives and interacts with the environment using high-dimensional sensory inputs (e.g. RGB images). We propose to learn both the policy and the perception module because (1) learning methods (e.g. neural networks)  have been shown effective in dealing with high-dimensional data and (2) a hand-crafted programmatic policy or perception module can be difficult to obtain when the agent's observation is high-dimensional. While we tried to provide this intuition at the beginning of Section 4, we apologize that it was not clear and have revised to paper to emphasize this.
>
> To further show that our framework can be extended to a high-dimensional state representation where a hand-crafted policy or perception module might not be easy to obtain, we performed an additional experiment where the perception module and the policy are trained on raw RGB inputs instead of symbolic state representation. The results suggest that our framework can utilize RGB inputs while maintaining similar performance (93.2% on the test set) and generalization ability (91.4% on the test-complex set). We have revised the paper to incorporate this finding. Please see Section E.5 for the details.
>
> Q: Section 4.4.1: I believe the explanation of the perception module would benefit from a concrete example.
>
> A: Thank you for the suggestion. We have added an example when explaining the perception module in the revision.

---

> ### Author Response · Authors · 2019-11-14
> **Response to Reviewer #1 (2/3)**
>
>
> Q: One aspect that I found most interesting in this paper is that the authors also let annotators map the given programs into natural language form. However, there is no discussion of these results. Similarly, there are interesting qualitative analyses in the appendix of the paper that I only stumbled upon by chance. I believe these should be referenced and a short summary should be integrated into the main part of the paper. I would particularly like to see a discussion of the limitations already in the main part of the paper.
>
> A: We completely agree that introducing the natural language counterpart of programs and providing some insights on learning from those annotated natural language instructions are informative.
> We have revised the paper to make it clear that there are some exemplary annotated natural language instructions (first column) and the programs (second column) presented in Figure 10. We will make the entire dataset publicly available for further research.
> Also, the analysis of learning from these annotated natural language instructions is provided in Section 5.4 and Section E.6.3. We analyzed how the length and diversity of instruction could affect various end-to-end learning models as well as how certain conditioning variables of a program could impact the performances.
> We would like to further clarify what the reviewer means by limitations so that we can incorporate this discussion into the paper.
>
> **Minor comments**
>
> Q: p1: I like the motivation of cooking recipes for work on program conditioned learning. There is in fact a paper (probably multiple) from the NLP community that I think could be cited here. The one that comes to my mind is: Malmaud, Jonathan, et al. "Cooking with semantics." Proceedings of the ACL 2014 Workshop on Semantic Parsing. 2014.
>
> A: Thank you for the suggestion. We have cited the mentioned paper together with a few related works along this line in the revised introduction.
>
> Q:  I agree with the argument that programs might be favored over natural language to specify goals as they are unambiguous. However, I think this can also be seen as a drawback. Natural language allows us to very efficiently share information, maybe sometimes information that is only disambiguated through observations in the environment. Another advantage is that natural language for instructing learning agents (like people) is abundant on the web, while programs are not.
>
> A: We completely agree that natural language is an efficient and expressive way of conveying ideas as humans can often leverage context, common sense, visual cues, etc. to disambiguate certain languages. In this work, we are mainly interested in advocating programs as a good alternative representation for instructing a learning agent.
> We also agree that the accessibility of natural languages is enjoyable. It is worth noting there has been growing interest in synthesizing programs from natural language (Lin et al., 2018; Raza et al., 2015; Desai et al., 2016) and other representations. On the other hand, or semantic parsing has been an active research area to bridge unstructured languages and structural formal languages (Wang et al., 2017).
>
> Q: "that leverages grammar" -"that leverages a grammar" and "we propose to utilize an precise" -"we propose to utilize a precise"
>
> A: Thank you for pointing out the errors. We have fixed them in the revised paper.
>
> Q: For learning from video demonstrations, an important prior work is Aytar, Yusuf, et al. "Playing hard exploration games by watching youtube." Advances in Neural Information Processing Systems. 2018.
>
> A: Thank you for pointing out these related works. We have discussed them in the revised paper.
>
> Q:  A deep learning program synthesis work prior to the ones mentioned here is Bošnjak, Matko, et al. "Programming with a differentiable forth interpreter." Proceedings of the 34th International Conference on Machine Learning-Volume 70. JMLR. org, 2017.
>
> A: Thank you for pointing out these related works. We have discussed them in the revised paper.

---

> ### Author Response · Authors · 2019-11-14
> **Response to Reviewer #1 (1/3)**
>
>
> We thank the reviewer for the thorough and constructive comments. Please find the response to your questions below:
>
> **Strengths**
>
> We appreciate the reviewer for recognizing our contributions.
>
> **Weaknesses**
>
> Q: One of the contributions of the paper is a modulation mechanism (Section 4.3) on the state features that incorporates a goal-conditioned policy. However, a very related approach has been proposed by Bahdanau, Dzmitry, et al. "Learning to Understand Goal Specifications by Modelling Reward." ICLR 2019. They introduced FILM layers that modulate the layers in a ConvNet conditioned on a goal representation. This should be discussed and compared to in the paper.
>
> A: To fuse the information from an input domain (e.g. an image) with another condition domain (e.g. a language query, image such as segmentation map, noise, etc.), a wide range of works have demonstrated the effectiveness of predicting affine transforms based on the condition to scale and bias the input in visual question answering (Perez et al., 2018; 2017), image synthesis (Almahairi et al., 2018; Karras et al., 2019; Park et al., 2019), style transfer (Dumoulin et al., 2017), recognition (Hu et al.,2018; Xie et al., 2018), reading comprehension (Dhingra et al., 2017), few-shot learning (Oreshkinet al., 2018; Ye et al., 2018; Lee & Choi, 2018), etc. Many of those works present an extensive ablation study to compare the learned modulation against traditional ways to merge the information from the input and condition domains.
> Recently, a few works have employed a similar learned modulation technique to reinforcement learning frameworks on learning to follow language instruction (Bahdanau et al. 2019) (as mentioned by the reviewer) and meta-reinforcement learning (Vuorio et al. 2019). However, a comprehensive ablation study that suggests fusing the information from the input domain (e.g. a state) and the condition domain (e.g. a goal or a task embedding) is lacking from these works. In this work, we conduct an ablation study that clearly shows the effectiveness of learning to modulate input features with a symbolically represented goal as well as present a number of modulation variations (i.e. modulating the fully-connected layers or the convolutional layers or both). We have incorporated this discussion in the revised paper in Section D.
>
> Q: I am surprised there is no comparison to other work that conditions on programs or hierarchical RL approaches. For example, the authors mention various works in Section 2 but fail to compare to them or at least explain why a comparison would not be possible.
> Another point of criticism is that the authors do not use an existing environment, but instead a Minecraft-inspired one similar to Andreas et al, Oh et al. and Sohn et al. This makes a comparison to prior work hard and I would like to understand in what way previous environments were inadequate for the research carried out here.
>
> A: We have added an extended discussion on hierarchical RL frameworks in Section C. We are not aware of any hierarchical RL frameworks that aim to fulfill a task represented by a written program that consists of control flows, perception, and actions.
> We would like to take the works mentioned here as an example. We believe it is not trivial to directly compare the proposed framework to the prior works including (Andreas et al. ICML 2017), (Oh et al. ICML 2017), (Sohn et al. NeurIPS 2018), etc. Both the models proposed in (Andreas et al. ICML 2017) and (Oh et al. ICML 2017) take a sequence of symbolically represented subtasks as input, and it is not clear how these models can deal with programs that consist of control flows, perceptions (i.e. conditions), and actions (i.e. subtasks).
> On the other hand, the framework proposed by (Sohn et al. NeurIPS 2018) requires a subtask graph describing a set of subtasks and their dependencies and aims to find the optimal subtask to execute. This is different from our problem formulation where the agent is asked to follow a given program/procedure.
> Therefore, given that it is not trivial to compare against the above method, we decided to implement our own environment that is similar to (Andreas et al. ICML 2017, Sohn et al. NeurIPS 2018) for the convenience. We have incorporated this discussion in the revised paper and made it clear why directly comparing our proposed framework to these works is not trivial.

---

### Official Review · AnonReviewer2 · 2019-10-23
**Official Blind Review #2**

**Rating:** 6

**Review:**

This paper presents a reinforcement learning agent that learns to execute tasks specified in a form of programs with an architecture consisting of three modules. The (fixed) interpreter module interprets the program, by issuing queries to a (pre-trained) vision module and giving goals to a policy module that executes them in the environment. The paper also introduces a policy modulation technique, with the goal of modulating the current state with the expected (symbolic) goal. The model is evaluated on a 2D approximation of Minecraft, where it outperforms a set of baselines. In addition, the authors modified the program dataset and re-expressed it in terms of natural language and showed that the baselines perform better with programs than with the natural language instructions.


Though I think the general idea of the paper is worth exploring, I find the concrete contribution of the paper a bit thin to the point that I am hesitant to recommend this paper for acceptance. Allow me to explain my objections:

First and foremost, this work is very close to work by Denil et al where the execute (in a differentiable fashion) programs in an RL agent. Their work does not have a discrete interpreter per-se, but it does have a differentiable execution of commands. The major difference between these two works would be that Denil et al do not have the vision module (they do mention learning from pixels as future work).
However, that is not entirely true. The model presented here uses a pretrained vision module, which by itself is not a problem and is used in related work [1], but this vision module does not operate on visual input but the symbolic representation of the map. A crucial thing here is that if all of a sudden we want to include a new object on the map, the model won’t be able to use some learned similarity since it would require introducing a new object (slice of the input), as it would should it have been trained on pixels. So technically speaking, this is not a vision module but a symbolic state processing module.

Then, the modulation mentioned in the paper does not seem particularly novel. Sure, the exact architecture of the model is probably unique, but the idea of modulating a state with a goal is not and has been seen in other work such as [2] and [3] among others. The paper does not mention why, for example, this modulation technique is useful and why any other similar architecture would not be as successful, nor does it mention related modulation techniques in other work.

A big issue I have with the evaluation in the paper is that I do not see the benefit of having the experiments with natural language at all. The focal point of the paper are agents able to execute tasks in the form of programs. The (though manually) generated natural language instructions from those same programs cannot even be used by the proposed agent as there is no natural language interpreter, so they are a dangling part of the paper which is there just to showcase that programs should be easier for baselines to learn to execute than natural language (the hypothesis would be that a simpler and more structured/formal language is easier to learn than the natural language). Hence the seq-LSTM results on language which are a tad lower than the results on programs are expected, though the performance of transformers is the opposite---they are better on language, and that is something that is puzzling and left unexplained, as well as the unexpectedly low performance of Tree-RNNs. One would expect them to perform a bit better than LSTMs, but that might be contingent on the size of the dataset more than the structure of the inputs. However, none of these curious findings have been explained.
Moreover, the comparison to baselines is not particularly fair as these baselines had to learn the symbolic state interpretation, whereas your model did not. You could have provided the same to the baselines for a better comparison.

In addition to that, 5.4.2 goes into detail of analysing the baselines, and ignoring the proposed model. Why didn’t you include the same statistics for your agent in Figure 5 and said subsubsection?

The paper is missing some notable related work:
- S.R.K. Branavan’s work (all but one cited language-instructed agent papers are post 2017) as well as [4]
- object-oriented and hierarchical RL
- [5], where they train the neural programmer interpreter from the final reward only, which brings NPI close to this work

Questions:
- Figure 5 - There’s a mark for Tree-RNN, but Tree-RNNs are not in any figure. Why didn’t you plot the performance of your model?
- The setup naturally induces a curriculum - how does it do that if the programs are randomly sampled?
- You say that your model learns to comprehending the program flow. I’m not sure I would agree with that because what your model learns is to execute single commands. From what it seems, the interpreter is the only part of the model (which is fixed btw) which sees the control flow, whereas the policy just executes singular commands. Did you mean something else by that statement?

Minor issues:
- You say twice interpreter (i.e. compiler). Given that they’re not the same, and you’re using an interpreter, I suggest omitting the word compiler.
- Figure 2 is lacking detail. There is no difference between r and i (both being positive integers) other than their descriptions, - operators agent[_], env and is_there lack parameters (non-terminal nodes), and where’s river, bridge, etc?

[1] COBRA: Data-Efficient Model-Based RL through Unsupervised Object Discovery and Curiosity-Driven Exploration
[2] FeUdal Networks for Hierarchical Reinforcement Learning
[3] Universal value function approximators
[4] Vogel et al Learning to follow navigational directions
[5] Improving the Universality and Learnability of Neural Programmer-Interpreters with Combinator Abstraction

**Experience Assessment:**

I have read many papers in this area.

**Review Assessment: Checking Correctness Of Derivations And Theory:**

I assessed the sensibility of the derivations and theory.

**Review Assessment: Checking Correctness Of Experiments:**

I assessed the sensibility of the experiments.

**Review Assessment: Thoroughness In Paper Reading:**

I read the paper at least twice and used my best judgement in assessing the paper.

---

> ### Author Response · Authors · 2019-11-14
> **Response to Reviewer #2 (4/4)**
>
>
> Q: 5.4.2 goes into detail of analysing the baselines, and ignoring the proposed model. Why didn’t you include the same statistics for your agent in Figure 5 and said subsubsection?
>
> A: The performance drop of evaluating our agent on the programs that are longer or have a higher number of control flows is less significant than the drop shown on the end-to-end learning baselines. We believe this is because our framework utilizes a rule-based parser (i.e. the program interpreter) which is not affected by longer or more complex tasks. Therefore, to analyze the cause of failures, we provided failure analyses in Section E.5.1 and Section E.5.2 detailing the failure rates and the average time cost of each subtask.
>
> Q: The paper is missing some notable related work:
> - S.R.K. Branavan’s work (all but one cited language-instructed agent papers are post 2017) as well as [4]
> - object-oriented and hierarchical RL
> - [5], where they train the neural programmer interpreter from the final reward only, which brings NPI close to this work
>
> A: Thanks for pointing out these relevant works. We have incorporated these works and some of their follow-up works in the revision.
>
> Q: Figure 5 - There’s a mark for Tree-RNN, but Tree-RNNs are not in any figure. Why didn’t you plot the performance of your model?
>
> A: Figure 5 presents all the combinations of models {Seq-LSTM, Transformer, Tree-RNN} and task representations {programs, natural language instructions}. While Seq-LSTM and Transformer can be applied to programs and natural language instructions, Tree-RNN requires the input representation with a tree structure and therefore is only evaluated on programs.
> The bottom-left subfigures of Figure 5(a) and 5(b) in the original paper both show the performance of Tree-RNN in dark blue. However, the dark red color was reserved for Tree-RNN applied to natural language instructions, which is not evaluated and therefore is not shown in this figure. We have removed the dark red color from the legend and we are sorry for the confusion.
>
> Q: The setup naturally induces a curriculum - how does it do that if the programs are randomly sampled?
>
> A: We did not explicitly set up a curriculum; instead, we always randomly sampled programs regardless of their difficulty. We found that this setup naturally induces a curriculum. At the beginning of the training, the policy first learns to solve programs that require a less number of subtasks but fails to complete the harder programs. Yet, completing simpler programs and obtaining the task completion reward still allows the model to obtain a better understanding of the subtasks, which eventually allows the policy to complete more complex programs. We have revised the paper to make it clear that we did not explicitly set up a curriculum.
>
> Q: You say that your model learns to comprehending the program flow. I’m not sure I would agree with that because what your model learns is to execute single commands. From what it seems, the interpreter is the only part of the model (which is fixed btw) which sees the control flow, whereas the policy just executes singular commands. Did you mean something else by that statement?
>
> A: We appreciate the comment. We have replaced the term “comprehend” with “execute” to make it is clear that our framework does not learn to read programs but utilizing a rule-based parser (i.e. the program interpreter).
>
> Q: You say twice interpreter (i.e. compiler). Given that they’re not the same, and you’re using an interpreter, I suggest omitting the word compiler.
>
> A: Thanks for the suggestion. We were intended to give an intuition when writing the paper but you are absolutely correct that they are not the same. We have removed it from the revised paper.
>
> Q: Figure 2 is lacking detail. There is no difference between r and i (both being positive integers) other than their descriptions, - operators agent[_], env and is_there lack parameters (non-terminal nodes), and where’s river, bridge, etc?
>
> A: We have fixed the DSL according to the suggestion. Specifically, we have made the following changes: (1) $r$ and $i$ were merged into $i$, which represents a constant, and (2) we added terrain ($u$) to represent different types of terrains.

---

> > ### Comment · AnonReviewer2 · 2019-11-14
> > **Reply**
> >
> > Thank you for very detailed replies and for updating the paper.
> >
> > - Paper vs Denil et al - there definitely are differences between your proposed model and theirs, but given how close the models are, I was hoping you’d explicitly state the difference. Thank you for doing that.
> > - Thank you for doing the RGB pixel experiment. Nice to see the model working on pixels. My main objection here was that calling a symbolic input visual is a bit misleading and would suggest emphasising that from the beginning of the paper.
> > - Thank you for the clarification of the differences between the modulated policy and the related work
> >
> > I do find the language experiments interesting, however they stick out of the paper given that the focus of almost all of the paper is the agent able to learn to execute programs. The program-executing agent is not applicable to natural language, nor can the program interpreter be pulled out of it (without decimating its accuracy?). In that sense, the program-executing agent presented in the first part of the paper seems like almost an oracle to the struggling baselines, which do not have a symbolic interpreter to use and have to ‘learn to interpret’, while at the same time solving the task.
> >
> > So, your points 1-4 absolutely hold ground, your findings are interesting, but they feel disconnected from the rest of the paper as they are not dependent on model which was the focus of the paper up until that point. In that sense the paper feels like combining two disparate things - very capable specialised program-executing agent, and the analysis of text vs programs on other more general models.
> >
> > I admit I might have been harsh though in my original review, and I apologise for that.
> >
> > Given your reply and the rest of the correspondence, I'm increasing my score.

---

> > > ### Author Response · Authors · 2019-11-15
> > > **Response to Reviewer #2**
> > >
> > >
> > > We are extremely grateful to the reviewer for acknowledging our revision and response as well as the prompt reply.
> > >
> > > *Visual input*
> > > We completely agree with this point and this is also why we did not call our perception module a visual/vision module in our paper. We thank the reviewer for inspiring us to conduct the RGB pixel experiment, which leads to this encouraging finding.
> > >
> > > *The connection to learning from natural language instructions*
> > > One of our main motivations for utilizing programs as a task representation is that we conjecture it is difficult for agents learning from simpler language instructions to generalize well to much more complex ones. Therefore, we believe it is important to verify this in our paper. We further demonstrate that even if the instructions follow a strict structure (i.e. programs), end-to-end learning models still struggle at generalizing to more complex tasks. This motivates us to develop our framework that leverages the structure of a program interpreter. We originally organized the paper such that the problem formulation is clear and we apologize that this motivation was not clear.
> > > While the key idea of this paper is to advocate utilizing programs as a task representation and propose a modular framework to address this problem, we look forward to future research that leverage the strengths of both natural languages and programs by bridging the gap between these two representations, such as synthesizing programs from natural language (Lin et al., 2018; Raza et al., 2015; Desai et al., 2016), semantic parsing that bridges unstructured languages and structural formal languages (Yu et al., 2018, Yin and Neubig 2018), and naturalizing program (Wang et al., 2017).
> > >
> > > *Our contributions and further potential concerns*
> > > We would like to emphasize our key contributions and verify them with the reviewer:
> > > - We propose to utilize imperative programs, structured in a formal language, as a precise and expressive way to specify tasks/desired procedures.
> > > - We devise a modular framework that leverages the program structure to execute programs as well as learn to perceive and interact with the environment.
> > > - We conduct an ablation study to show the effectiveness of the learned modulation mechanism in learning a multitask/goal-condition policy.
> > > - We empirically demonstrate the poor generalization ability of end-to-end learning frameworks that learn from natural language instructions or programs.
> > > - Our proposed framework outperforms the end-to-end learning on both the performance and generalization ability.
> > > Please kindly let us know if there are any further concerns or missing experimental results that potentially prevent you from accepting this submission. We would be more than happy to address them if time allows. Thank you very much for all your detailed feedback and the time you put into helping us to improve our submission.
> > >
> > > *Reference*
> > > Lin et al. "NL2Bash: A corpus and semantic parser for natural language interface to the linux operating system" LREC 2018
> > > Desai et al. “Program synthesis using natural language” ICSE 2016
> > > Raza et al. “Compositional program synthesis from natural language and examples” IJCAI 2015
> > > Yu et al. “Spider: A large-scale human-labeled dataset for complex and cross-domain semantic parsing and text-to-sql task” EMNLP 2018
> > > Yin and Neubig “TRANX: A Transition-based Neural Abstract Syntax Parser for Semantic Parsing and Code Generation” EMNLP 2018
> > > Wang et al. “Naturalizing a Programming Language via Interactive Learning” ACL 2017

---

> ### Author Response · Authors · 2019-11-14
> **Response to Reviewer #2 (3/4)**
>
>
> Q: A big issue I have with the evaluation in the paper is that I do not see the benefit of having the experiments with natural language at all. The focal point of the paper are agents able to execute tasks in the form of programs. The (though manually) generated natural language instructions from those same programs cannot even be used by the proposed agent as there is no natural language interpreter, so they are a dangling part of the paper which is there just to showcase that programs should be easier for baselines to learn to execute than natural language (the hypothesis would be that a simpler and more structured/formal language is easier to learn than the natural language). Hence the seq-LSTM results on language which are a tad lower than the results on programs are expected, though the performance of transformers is the opposite---they are better on language, and that is something that is puzzling and left unexplained, as well as the unexpectedly low performance of Tree-RNNs. One would expect them to perform a bit better than LSTMs, but that might be contingent on the size of the dataset more than the structure of the inputs. However, none of these curious findings have been explained.
>
> A: Natural language is one of the most straightforward and efficient ways of specifying desired tasks. Therefore, developing an agent that can follow natural language instructions has been an active research area. While in this work, we aim to demonstrate that program is a good alternative for instructing agents, we believe our paper provides some insights on (1) comparing models learning from two instruction representations, (2) examining models learning from longer and more diverse (i.e. more branches) natural language instructions that are usually not concerned in most prior works, (3) comparing network architectures for learning from natural language instructions, and (4) the performance drop shown by evaluating models on complex natural language instructions when they only learn from simpler ones. We would like to believe that this will be informative to the research community that focuses on learning from language.
>
> We provide our analysis and hypothesis for the experimental results in Section 5.4 (Section 5.4.1 and Section 5.4.2) and the captions of Figure 5. Specifically, we show that programs with a strict structure may lead to a performance mismatch when using the Transformer. On the other hand, the Transformer may find an easier way of languages as in some cases understanding the task given by language does not require strict sequential processing.
>
> As can be seen from Table 1, Tree-RNN does achieve the best generalization with programs. The relatively poor performance compared to Seq-LSTM may be attributed to the fact that Tree-RNN needs to learn to understand the order of the children's merging operation is crucial to the overall program comprehension. We hypothesize that with larger and more diverse program sets, the Tree-RNN can achieve better performance.
>
> Q: The comparison to baselines is not particularly fair as these baselines had to learn the symbolic state interpretation, whereas your model did not. You could have provided the same to the baselines for a better comparison.
>
> A: You are absolutely correct. We apologize for missing the details of how we aimed to make the comparison fair.
> Since we train our perception module using ground truth information, we actually did provide the ground truth perception information to all our baselines yet failed to elaborate this in the original paper. Specifically, at every time step, we feed the ground truth perception (\ie the answer to the queries such as env[Gold]>0 and is_there[River]) to the baseline models. Therefore, the baseline models can learn to utilize this ground truth information to infer the desired subtasks.
> During testing time, the baseline models can still access to all this ground truth perception information, even though it is usually not possible in practice. On the contrary, during testing time, our perception module predicts the answer to given queries and the performance of the whole framework depends on the predicted answers.

---

> ### Author Response · Authors · 2019-11-14
> **Response to Reviewer #2 (2/4)**
>
>
> Q: Then, the modulation mentioned in the paper does not seem particularly novel. Sure, the exact architecture of the model is probably unique, but the idea of modulating a state with a goal is not and has been seen in other work such as [2] and [3] among others. The paper does not mention why, for example, this modulation technique is useful and why any other similar architecture would not be as successful, nor does it mention related modulation techniques in other work.
>
> A: Our policy modulation aims to use a given goal representation to predict a set of affine transformation parameters that are used to scale and bias the encoded state features. We believe this allows the policy to effectively attend to relevant objects (e.g. build_briedge() -> river, mine[gold] -> gold). The two works mentioned by the reviewer do not predict affine transforms from goals to modulate state features. Specifically, feUdal networks merge a state embedding with a goal embedding via an element-wise product and universal value function approximators form the merge features by matrix factorization.
>
> We believe the learned modulation mechanism would be effective due to its success in a wide range of applications, including visual question answering (Perez et al., 2018; 2017), image synthesis (Almahairi et al., 2018; Karras et al., 2019; Park et al., 2019), style transfer (Dumoulin et al., 2017), recognition (Hu et al.,2018; Xie et al., 2018), reading comprehension (Dhingra et al., 2017), few-shot learning (Oreshkinet al., 2018; Ye et al., 2018; Lee & Choi, 2018), RL (Bahdanau et al. 2019, Vuorio et al. 2019), etc. However,  while learned modulation has been shown effective in some applications, there has not been a comprehensive ablation study showing that this technique is effective in RL. Therefore, our work adapts this technique and presents an ablation study showing its effectiveness. This has been included in the revised paper (Section D).
>
> Moreover, as suggested by the reviewer, we experimented with merging features via an element-wise product (feUdal networks) or an element-wise summation in a feature space, and both variations show similar performance compared to the “concat” model. Due to the limited timeline, we only conduct experiments with two random seeds for each variation. We will revise the paper once all the experiments are completed.

---

> ### Author Response · Authors · 2019-11-14
> **Response to Reviewer #2 (1/4)**
>
>
> We thank the reviewer for the thorough and constructive comments. Please find the response to your questions below:
>
> Q: First and foremost, this work is very close to work by Denil et al where the execute (in a differentiable fashion) programs in an RL agent. Their work does not have a discrete interpreter per-se, but it does have a differentiable execution of commands. The major difference between these two works would be that Denil et al do not have the vision module (they do mention learning from pixels as future work).
> However, that is not entirely true. The model presented here uses a pretrained vision module, which by itself is not a problem and is used in related work [1], but this vision module does not operate on visual input but the symbolic representation of the map.
>
> A: We would like to emphasize that our work differs from programmable agents (Denil et al., 2017) in several different ways. We describe the major differences as follows:
>
> - Denil et al. (2017) concern declarative programs which specify "what" to be computed (e.g. an end state such as a target object in a reaching task). However, the programs considered in our work are imperative, which "how" this is to be computed (i.e. a procedure). Therefore, given this significant difference in problem formulation, our proposed framework and programmable agents cannot be easily extended to replace one another.
> - Denil et al. (2017) consider only one-liner programs that contain only AND, OR, and object attributes. On the other hand, we consider programs that are much longer and describe more complex procedures.
> - While Denil et al. (2017) aim to generalize to novel combinations of object attributes, our work is mainly interested in generalizing to more complex tasks (i.e. programs) by leveraging the structure of programs.
>
> With the differences between programmable agents and our work in motivation, problem formulations, proposed methods, and contributions, we believe our work is novel and valuable. We have included this discussion in Section C.
>
> Q: A crucial thing here is that if all of a sudden we want to include a new object on the map, the model won’t be able to use some learned similarity since it would require introducing a new object (slice of the input), as it would should it have been trained on pixels. So technically speaking, this is not a vision module but a symbolic state processing module.
>
> A: We appreciate the reviewer for pointing out that the original setup does not support adding new objects. To verify if our framework can be extended to using RGB image as inputs and therefore can potentially generalize to new objects, we performed an additional experiment where the perception module and the policy are trained on raw RGB inputs instead of the symbolic state representation. The results suggest that our framework can utilize RGB inputs while maintaining similar performance (93.2% on the test set) and generalization ability (91.4% on the test-complex set). We have revised the paper to incorporate this finding. Please see Section E.5 for the details.

---

### Official Review · AnonReviewer3 · 2019-10-27
**Official Blind Review #3**

**Rating:** 8

**Review:**

This paper provides a method for instructing an agent using programs as input instructions, so that the agent should learn to contextually execute this program in a specified environment, learning to generalise as needed from perception, and to satisfy concerns that in the language of planning would be called monitoring and execution. The authors provide a method that breaks this problem down into one of interpreting the program (which is crafted separately as a compiler that benefits from a DSL), learning to identify contextual features and then adapting and applying the policy.

The arguments in this paper are well made but the paper would benefit from better clarifying several points:

1. To start at the very beginning, the authors begin in the first page by giving the impression that the agent has gone directly from an NL instruction and otherwise uninterpreted sensory input to a solution, in the spirit of typical end to end systems, whereas what the authors are proposing is a very different and more pragmatic approach wherein the interpretation of the task is handled prior to learning, so that learning is only applied to smaller subproblems. This could be made clearer in the introduction.

2. In particular, it was unclear how the DSL comes about and what restrictions it places on the problem. The DSL will clearly have an influence because a very different task from MineCraft, say, robot manipulation, would have quite different needs of sensor-driven control and hence the information flows (specifically, the separation between goal identification, context perception and motion execution) would be different. What one puts into the DSL will significantly influence how well the overall framework performs (e.g., the ability to crisply ask is_there is powerful). Have the authors systematically explored this axis of design? Can we hear more in the setup about this?

3. The influence of the domain is once again seen in the modulation mechanism for the goal and the way in which affine transformations enable generalisation. This is of course sensible in 2D spatial navigation but may be less straight forward in other decision making contexts. The authors have been clear enough about what they have done, but I would have found it interesting to understand how much we should expect this particular concept to stretch and where its limitations become more apparent - perhaps in the discussion.

Overall, this is good work and the writing is clear with suitable references. I would note that the authors are revisiting concerns well studied in the planning literature. While the authors do acknowledge HAMs and so on from the HRL literature, they'd make the paper stronger by also tracing the roots of some of these ideas into the rest of planning.

I'll end this review by asking about the relationships between NL instructions and the formal programs. In some domains, the number of realisable programs that map to an ambiguous NL instruction can be large. Equally, not all lay users can write good programs. So, it is worth noting this gap and making clear that this paper does not really address this bit of the overall problem.

**Experience Assessment:**

I have published one or two papers in this area.

**Review Assessment: Checking Correctness Of Derivations And Theory:**

N/A

**Review Assessment: Checking Correctness Of Experiments:**

I assessed the sensibility of the experiments.

**Review Assessment: Thoroughness In Paper Reading:**

I read the paper thoroughly.

---

> ### Author Response · Authors · 2019-11-14
> **Response to Reviewer #3 (2/2)**
>
>
> Q: Overall, this is good work and the writing is clear with suitable references. I would note that the authors are revisiting concerns well studied in the planning literature. While the authors do acknowledge HAMs and so on from the HRL literature, they'd make the paper stronger by also tracing the roots of some of these ideas into the rest of planning.
>
> A: Classical symbolic planning (Ghallab et al., 2004) considers a planning problem contains an initial state, a goal, and a set of operators where the agent should act accordingly. Each operator consists of the name of the operator (and necessary arguments), a precondition for this operator, and the effect on the environment if this operator is executed.
>
> As a whole, our framework shares many similarities to classical symbolic planning. Our policy (action module) which conditions on a symbolically represented goal is similar to an operator, as both the policy and an operator interacts with the environment to fulfill the given task. On the other hand, perceptions (conditions) in our DSL and preconditions both require perceiving environments (e.g. the presence/absence of objects). However, perceptions differ from preconditions in that perceptions determine which branches of a program should be taken while preconditions determine if an operator/action/skill is applicable.
>
> Q: I'll end this review by asking about the relationships between NL instructions and the formal programs. In some domains, the number of realisable programs that map to an ambiguous NL instruction can be large. Equally, not all lay users can write good programs. So, it is worth noting this gap and making clear that this paper does not really address this bit of the overall problem.
>
> A: As pointed out by the reviewer, there is a trade-off between the accessibility to the instructions (NL or programs) and the difficulty of learning from them. In this work, we are mainly interested in advocating programs as a good alternative representation for instructing an agent especially when the tasks are diverse (i.e. more branches).
> We also agree that the accessibility of natural languages is enjoyable. It is worth noting there has been growing interest in synthesizing programs from natural language (Lin et al., 2018; Raza et al., 2015; Desai et al., 2016) and other representations (e.g. images, strings, videos/execution traces. etc.). We would like to believe that the difficulty of accessing programs could be alleviated. On the other hand, semantic parsing has been an active research area to bridge unstructured languages and structural formal languages (Wang et al., 2017).

---

> > ### Comment · AnonReviewer3 · 2019-11-15
> > **Thanks for detailed clarifications**
> >
> > Re. last point on NL vs programs, I do acknowledge that there is work on going from NL to programs. However, I'd expect that having that as a front end to your current framework would either lower performance of add complexities due to redundancy in interpretation. Meanwhile, having a direct program interface makes it less intuitive to the entirely lay user.
> > That said, I am not saying this takes away from your paper, but being transparent at the outset about this point and acknowledging this point eliminates scope for doubt.

---

> > > ### Author Response · Authors · 2019-11-15
> > > **Response to Reviewer #3: clarification of NL vs programs**
> > >
> > > We sincerely thank the reviewer for the prompt response and the clarification. We completely agree that we should make it clear that addressing the accessibility of programs or converting a natural language instruction to a more structural form is beyond the scope of this work. We have revised our paper and incorporated this discussion in Section C. Please let us know if the reviewer believes it would be better to mention this point somewhere else in the paper so that we can further revise the paper accordingly.

---

> ### Author Response · Authors · 2019-11-14
> **Response to Reviewer #3 (1/2)**
>
>
> We thank the reviewer for the thorough and constructive comments. Please find the response to your questions below:
>
> Q: The authors begin in the first page by giving the impression that the agent has gone directly from an NL instruction and otherwise uninterpreted sensory input to a solution, in the spirit of typical end to end systems, whereas what the authors are proposing is a very different and more pragmatic approach wherein the interpretation of the task is handled prior to learning, so that learning is only applied to smaller subproblems. This could be made clearer in the introduction.
>
> A: We revised the first paragraph of the introduction to make it clear that we propose a modular framework instead of an end-to-end system.
>
> Q: It was unclear how the DSL comes about and what restrictions it places on the problem. The DSL will clearly have an influence because a very different task from MineCraft, say, robot manipulation, would have quite different needs of sensor-driven control and hence the information flows (specifically, the separation between goal identification, context perception and motion execution) would be different. What one puts into the DSL will significantly influence how well the overall framework performs (e.g., the ability to crisply ask is_there is powerful). Have the authors systematically explored this axis of design? Can we hear more in the setup about this?
>
> A: You are absolutely correct. Different domains require different DSLs. We aim to design our DSL by following a design principle that would potentially allow us to easily adapt our DSL to a different domain. Specifically, we develop a DSL design principle that considers a general setting where an agent can perceive and interact with the environment to fulfill some tasks. Accordingly, our DSL consists of control flows, perceptions, and actions. While control flows are domain-independent, perceptions and actions can be designed based on the domain of interest, which would require certain expertise and domain knowledge.
> As noted by the reviewer, the DSL design does affect learning and performance. We aimed to design our DSL that is (1) intuitive: the actions and perceptions are intuitively align with human common sense, (2) modular: actions are reasonably distinct and can be used to compose more complex behaviors, and (3) hierarchical: a proper level of abstraction that enables describing long-horizon tasks. We have incorporated this discussion in Section B.
>
> Q: The influence of the domain is once again seen in the modulation mechanism for the goal and the way in which affine transformations enable generalisation. This is of course sensible in 2D spatial navigation but may be less straight forward in other decision making contexts. The authors have been clear enough about what they have done, but I would have found it interesting to understand how much we should expect this particular concept to stretch and where its limitations become more apparent - perhaps in the discussion.
>
> A: The learned modulation has been shown effective on a wide range of applications, including visual question answering (Perez et al., 2018; 2017), image synthesis (Almahairi et al., 2018; Karras et al., 2019; Park et al., 2019), style transfer (Dumoulin et al., 2017), recognition (Hu et al.,2018; Xie et al., 2018), reading comprehension (Dhingra et al., 2017), few-shot learning (Oreshkinet al., 2018; Ye et al., 2018; Lee & Choi, 2018), RL (Bahdanau et al. 2019, Vuorio et al. 2019), etc.
> Therefore, we would like to believe this modulation mechanism is general and is not limited to our environment. We have incorporated this discussion in the revised paper in Section D. However, we would like to further clarify what the reviewer means by “other decision-making contexts” here so that we can acknowledge potential limitations in the paper.

---

> > ### Comment · AnonReviewer3 · 2019-11-15
> > **Not all tasks are navigation to goal**
> >
> > re. the last question about the modulation mechanism, I take your point that there are many domains where this has worked but these are all domains differing in the perceptual presentation of what is at its core a 2D navigation task or similar. There are many tasks where the control side is more intricate, e.g., robot manipulation, say, where the specification is not just to change the goal but also other path properties.

---

> > > ### Author Response · Authors · 2019-11-15
> > > **Response to Reviewer #3: modulation in different domains**
> > >
> > > We appreciate the reviewer for clarifying this point. The reviewer is correct that it is possible that this learned modulation mechanism might not be effective when dealing with more complex domains such as robot manipulation. We have incorporated this discussion in Section D to make it is clear that we specifically verified the effectiveness of this mechanism in navigational tasks in a 2D Minecraft environment.

---

### Author Response · Authors · 2019-11-14
**Paper Revision**


We would like to sincerely thank all the reviewers for their thorough and constructive comments. We have revised our paper to incorporate them. The major changes are summarized as follows:

(1) [Introduction] We revised the introduction to make it is clear that this paper proposes a modular framework instead of an end-to-end learning system (R3). We also included a few papers that aim to learn to cook by following instructions (R1).

(2) [Related Work] We thank the reviewers for pointing out many relevant works and we have revised the related work section to include the papers in the following directions:
*Learning from natural language instructions* (Reviewer #2)
*Language to program synthesis* (Reviewer #1, Reviewer #3)
*Learning from demonstrations* (Reviewer #1)
*Neural program induction* (Reviewer #2)
*Neural program synthesis* (Reviewer #1)
*Planning* (Reviewer #3)

(3) [Policy modulation] We have included an additional section in the appendix (Section D) that presents a comprehensive survey on learned modulation and explicitly states our contribution.

(4) [RGB state representation] We further experimented with our proposed framework with the perception module and the policy to learning from high dimensional state input (RGB image) instead of the symbolic representation to address the questions asked by Reviewer #1 and Reviewer #2. The result suggests that our framework can still achieve good performance and exhibit a small generalization gap, which is presented in Section E.5.

(5) [DSL design principle] (Reviewer #3) We have included a discussion in Section B, explaining how we designed our DSL and how the principle that we followed could be extended to other domains.

(6) [Baseline perceptions] (Reviewer #2) We described how we augmented the end-to-end learning baselines with ground truth perceptions in Section E.2.

(7) [Hierarchical RL and multitask RL] (Reviewer #1) We have included an extended discussion on hierarchical RL and multitask RL in Section C.

(8) [Clarification] We have incorporated the comments to make the paper clear, including:
Revised the legend of Figure 5. (Reviewer #2)
Removed the term compiler. (Reviewer #2)
Replaced “comprehending programs” with “executing programs”. (Reviewer #2)
Fixed the DSL (Figure 2). (Reviewer #2)
Made it clear that we did not explicitly set up a curriculum. (Reviewer #2)
Added an example when explaining the perception module. (Reviewer #1)

(9) [Grammatical errors] (Reviewer #1) We have fixed the grammatical errors in the revision.

---

### Public Comment · ~Jingtong_Zhao1 · 2020-11-16
**Interesting Idea**

The idea is pretty interesting.

---

### Public Comment · ~Jizhou_Wu1 · 2022-10-27
**Not quite understand how the programs are generated.**

Is there a program generator that randomly samples components (perception, items, operators, etc) from DSL and then assembles them? If so, how it is optimized?
In the experiment part, "During training, we randomly sample programs from the training set as well as randomly sample an environment state to execute the program interpreter". What does "an environment state" mean? Different states in an environment? There are thousands of programs in the training set and test set, I am not sure whether they're all correspond to the same environment/task?

---

### Decision · Program_Chairs · 2019-12-19

**Decision:**

Accept (Spotlight)

**Comment:**

This paper provides a fascinating hybridization approach to incorporating programs as priors over policies which are then refined using deep RL. The reviewers were, at the end of the discussion, all in favour of acceptance (with the majority strongly in favour). An excellent paper I hope to see included in the conference.